# TRAINING-FREE EXPONENTIAL CONTEXT EXTENSION VIA CASCADING KV CACHE

[1]**Jeffrey Willette, **[1]**Heejun Lee, **[1,2]**Youngwan Lee**
[1]KAIST AI, [2]ETRI, South Korea
{jwillette,ainl}@kaist.ac.kr,yw.lee@etri.re.kr

**Myeongjae Jeon**
POSTECH, South Korea
mj.jeon@postech.ac.kr

**Sung Ju Hwang**
KAIST AI, Deepauto.ai, South Korea
sungju.hwang@kaist.ac.kr

## ABSTRACT

The transformer's context window is vital for tasks such as few-shot learning and conditional generation as it preserves previous tokens for active memory. However, as the context lengths increase, the computational costs grow quadratically, hindering the deployment of large language models (LLMs) in real-world, long sequence scenarios. Although some recent key-value caching (KV Cache) methods offer linear inference complexity, they naively manage the stored context, prematurely evicting tokens and losing valuable information. Moreover, they lack an optimized prefill/prompt stage strategy, resulting in higher latency than even quadratic attention for realistic context sizes. In response, we introduce a novel mechanism that leverages cascading sub-cache buffers to selectively retain the most relevant tokens, enabling the model to maintain longer context histories without increasing the cache size. Our approach outperforms linear caching baselines across key benchmarks, including streaming perplexity, question answering, book summarization, and passkey retrieval, where it retains better retrieval accuracy at 1M tokens after four doublings of the cache size of 65K. Additionally, our method reduces prefill stage latency by a factor of 6.8 when compared to flash attention on 1M tokens. These innovations not only enhance the computational efficiency of LLMs but also pave the way for their effective deployment in resource-constrained environments, enabling large-scale, real-time applications with significantly reduced latency.

## 1 INTRODUCTION

Large language models (LLMs) have become indispensable in a wide range of applications, from natural language processing to AI-driven content generation. However, their deployment is often hindered by the significant computational resources required, particularly during the quadratic attention operation in inference. Despite recent advancements such as Flash Attention 2 (Dao, 2023), which reduce memory overhead, the quadratic growth of latency and compute costs with input size remains a burden, especially when processing long input sequences. This challenge is felt in streaming applications, where high latency directly impacts user experience and operational costs.

Existing methods, such as sliding window approaches (Beltagy et al., 2020; Jiang et al., 2023), attempt to manage long sequences but impose a static limit on the model's ability to retain context due to the fixed window size. As a result, valuable tokens are naively discarded once they fall outside the window, leading to irreversible loss of context. Although some recent meth-

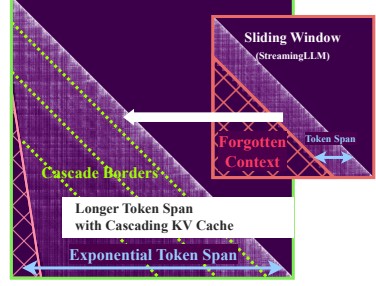

**Cascading KV Cache (Ours)**

Figure 1: Attention matrices from Streaming LLM (Xiao et al., 2023) and Cascading KV Cache (Ours), **both with the same total cache size**.

ods have tried to stabilize this process by preserving a small number of initial tokens (Xiao et al., 2023) (Figure 3 top), they still operate within a static framework with a naive eviction policy that does not account for the relative importance of tokens within the sequence. Additionally, the linear inference procedure from Xiao et al. (2023) processes a single token per step, which is impractical to apply during the prompting stage when many tokens need to be processed together in parallel.

Our work addresses these critical limitations by proposing a novel linear inference approach that extends the effective context length of sliding windows without increasing computational complexity or requiring additional training. We introduce a dynamic caching mechanism that organizes the key-value (KV) cache into cascading sub-caches, each designed to selectively retain tokens based on their importance. As opposed to a fixed sliding window that only evicts tokens at the end, our method offers multiple eviction routes before reaching the end, while intelligently preserving older tokens, which are likely to play a crucial role in future predictions.

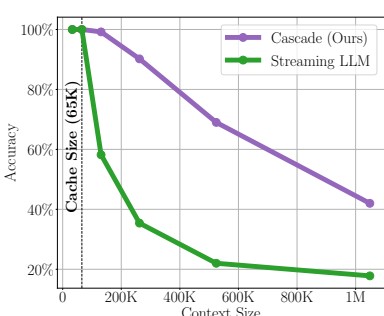

Figure 2: Passkey accuracy up to 1M tokens given a cache size of 65K. Our Cascading cache maintains higher accuracy even after four doublings of the context length.

To demonstrate the effectiveness of our method, we present two compelling examples: First, we visualize the attention matrix on the PG19 test set, showing how our cascading KV cache retains context far beyond a static sliding window while maintaining the same total cache size (Figure 1). Second, we provide an example on passkey retrieval up to 1M tokens given a total cache size of 65K (Figure 2), illustrating that our method consistently outperforms static window attention in terms of retrieval accuracy, maintaining superior performance over prior work even after **four doublings** of the context size. These results underscore the potential of our approach to transform the efficiency and accuracy of long-context LLMs in both research and real-world applications. Our contributions are as follows:

- We introduce a simple yet powerful training-free linear attention modification to sliding window attention on pretrained quadratic transformers that selectively retains important tokens, significantly extending the effective context length.

- Given the same total KV cache size, our method delivers substantial improvements of 12.13% average in long context benchmarks (LongBench), 0.4% in streaming perplexity (PG19), 4.48% in Book Summarization, and increases passkey retrieval accuracy by 24 percentage points (pp) at 1M token after four doublings of the context size and 18pp higher accuracy after 5 doublings of the context size.

- We provide a linear prefill strategy that avoids the restrictive quadratic prompt complexity of previous works. Our strategy reduces latency on 1M tokens by a factor of 6.8 compared to Flash Attention 2.

- We provide an efficient implementation[1] of our KV cache, which achieves a more than two order of magnitude speedup over Streaming LLM.

## 2 RELATED WORK

Previous research has explored sparse attention patterns to reduce computational costs, such as in BigBird (Beltagy et al., 2020), where a sliding window with random sparse context is applied. While effective in reducing time complexity, this approach relies on uninformative random attention patterns that lack adaptability. Linear attention mechanisms, including locality-sensitive hashing (Kitaev et al., 2020), kernel-based approximations (Choromanski et al., 2020), attention matrix compression (Lee et al., 2023), and token compression techniques (Munkhdalai et al., 2024; Kim et al., 2023; Mohtashami & Jaggi, 2023), offer promising alternatives but often require extensive retraining, limiting their broad applicability to existing models.

Our work builds on insights from Streaming LLM (Xiao et al., 2023), which identified a key phenomenon in transformers: as features progress through layers, attention scores often concentrate

---

[1] https://github.com/jeffwillette/cascading_kv_cache

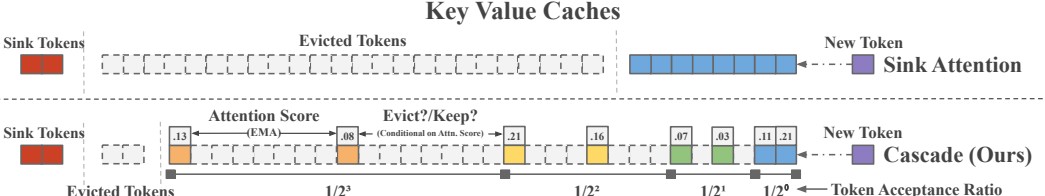

Figure 3: Comparison of Streaming LLM (Xiao et al., 2023) and Cascading Cache (Ours). **Top**: Streaming LLM stores **fixed sink tokens (red)** along with a **sliding window** of $N$ recent tokens. **Bottom**: Our method segments the cache into smaller cascading sub-caches, where each successive sub-cache conditionally accepts a fraction of tokens based on the magnitude of past attention scores. This simple technique allows for important tokens to remain in the cache for a longer time instead of being naively evicted too early. Conversely, superfluous tokens may be evicted before reaching the end of the cache, allowing for an intelligent eviction strategy.

probability mass on the initial tokens, referred to as "sink tokens." This phenomenon presents a challenge in traditional fixed sliding windows, as evicting sink tokens from the cache disrupts the attention score distribution, forcing the model to redistribute probability mass onto other unexpected tokens. Xiao et al. (2023) demonstrated that this distribution shift leads to significant performance degradation unless the sink tokens are retained. We extend this insight by introducing an efficient, scalable method that increases the effective context window while maintaining a static cache size, improving throughput during the prompt stage, and enhancing performance on long-sequence tasks.

Subsequent approaches, such as SnapKV (Li et al., 2024) and H2O (Zhang et al., 2024b), have introduced score-based KV cache eviction policies, but these rely on quadratic prompts, with cache compression limited only to the decoding phase. These methods lack a linear prefill strategy, which results in the prefill attention operation remaining quadratic. Our method diverges from these previous approaches by offering a fast linear prefill strategy that is compatible with a cache that selectively retains important tokens through dynamic sparsity. This makes our approach not only more adaptive but also significantly more efficient, providing a practical path to extend transformer models' context length with linear complexity and without incurring the high costs of retraining.

## 3 METHOD

**Notation.** We use the common notation of a boldface lowercase letter to denote a vector $\boldsymbol{x}$ and a boldface uppercase letter to denote a matrix $\boldsymbol{X}$. A superscript $(l)$ denotes that an object belongs to layer $l$, where $l \in [1, 2, \ldots, L]$. To simplify notation for attention, we omit the head dimension, output projections, and multi-layer perceptron (MLP) transformations, please see Vaswani et al. (2017) for an overview regarding those topics. We refer to a generic cache as $\boldsymbol{C}$, using subscripts $\boldsymbol{C}_K$ and $\boldsymbol{C}_V$ to refer to key and value caches, respectively.

**Attention.** Let $S \in \mathbb{N}$ represent a token sequence length, where each token at layer $l$ is represented by a vector $\boldsymbol{x}_i^{(l)} \in \mathbb{R}^d$. The collection of tokens in the sequence can be represented as a matrix $\boldsymbol{X}^{(l)} \in \mathbb{R}^{S \times d}$. With $\sigma$ being the softmax function, and different learnable query, key, and value matrices $\boldsymbol{Q}^{(l)}, \boldsymbol{K}^{(l)}, \boldsymbol{V}^{(l)} \in \mathbb{R}^{d \times d}$ the standard attention operation is as follows:

$$\boldsymbol{X}^{(l+1)} = \sigma \left( \frac{1}{\sqrt{d}} \left( \boldsymbol{X}^{(l)} \boldsymbol{Q}^{(l)} \right) \left( \boldsymbol{X}^{(l)} \boldsymbol{K}^{(l)} \right)^\top \right) \boldsymbol{X}^{(l)} \boldsymbol{V}^{(l)} \in \mathbb{R}^{S \times d} \tag{1}$$

**Key-Value (KV) Caching.** During LLM inference, a single token is generated at each time step. Combined with a causality condition such that token $\boldsymbol{x}_i$ cannot influence $\boldsymbol{x}_j$ iff $i > j$, it becomes more efficient to cache the $K^{(l)}$ and $V^{(l)}$ projected tokens in each layer $l$ rather than recomputing the full set of key, and value projections at each generation time step. Specifically, with $\cup$ representing a concatenation operation along the $S$ dimension, and each cache $\boldsymbol{C} \in \mathbb{R}^{S \times d}$, the calculation of the attention operation for the current token $\boldsymbol{x}_j$ during inference becomes,

$$\boldsymbol{x}_j^{(l+1)} = \sigma \left( \frac{1}{\sqrt{d}} \left( \boldsymbol{Q}^{(l)} \boldsymbol{x}_j^{(l)} \right)^\top \left( \boldsymbol{C}_K^{(l)} \cup \left( \boldsymbol{K}^{(l)} \boldsymbol{x}_j^{(l)} \right) \right)^\top \right) \left( \boldsymbol{C}_V^{(l)} \cup \left( \boldsymbol{V}^{(l)} \boldsymbol{x}_j^{(l)} \right) \right) \in \mathbb{R}^d \tag{2}$$

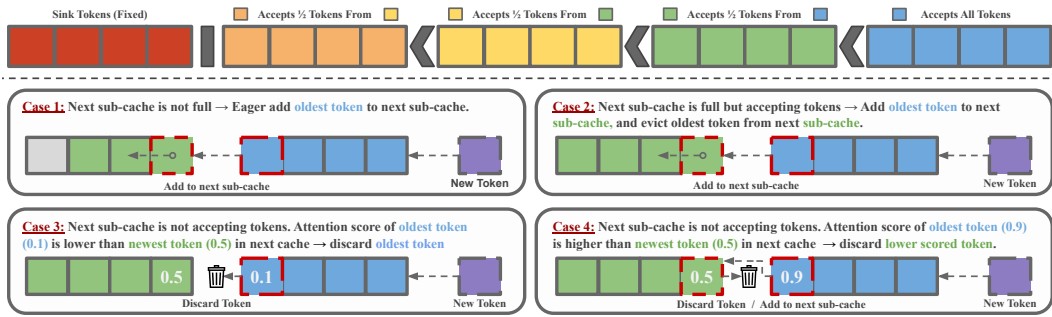

**Figure 4: Top:** Each successive sub-cache window accepts a fraction of tokens evicted from the previous sub-cache. **Bottom:** At the boundaries between sub-caches, there are four possible cases where our method takes a different conditional action, creating a dynamic attention pattern. Circular buffers are not depicted for simplicity of visualization.

After the concatenation operation, $\boldsymbol{x}_j^{(l)} \boldsymbol{K}^{(l)}$ and $\boldsymbol{x}_j^{(l)} \boldsymbol{V}^{(l)}$ are considered to be added to the respective $\boldsymbol{C}_K$ and $\boldsymbol{C}_V$ caches for the next iteration in the sequence.

Sliding window attention, as used by Streaming LLM, treats the KV cache as a fixed size buffer which must evict tokens from the cache when the cache reaches its storage limit. Considering a cache which is fully populated (denoted by an overbar $\overline{\boldsymbol{C}}_K^{(t)}$ and omitting the layer index $l$), when a new token comes in, the cache must drop the oldest token in order to make space for the new token such that,

$$\overline{\boldsymbol{C}}_K^{(t+1)} = \overline{\boldsymbol{C}}_K^{(t)} \cup \boldsymbol{K}\boldsymbol{x}_j = [\{\boldsymbol{K}\boldsymbol{x}_i, \boldsymbol{K}\boldsymbol{x}_{i+1}, \ldots, \boldsymbol{K}\boldsymbol{x}_{j-1}\} \cup \{\boldsymbol{K}\boldsymbol{x}_j\}] \setminus \{\boldsymbol{K}\boldsymbol{x}_i\}. \qquad (3)$$

The only difference between standard sliding window attention and sink cache of Streaming LLM (Xiao et al., 2023) is that the sink cache perpetually retains the first $\alpha$ tokens in the sequence such that if a full cache starts from the first token $\boldsymbol{x}_1 \boldsymbol{K}$, it is not token $\boldsymbol{x}_1 \boldsymbol{K}$ which is evicted, but token $\boldsymbol{x}_{1+\alpha} \boldsymbol{K}$, with $\alpha \in \mathbb{N}$ being a fixed hyperparameter. Both sliding window attention and sink cache have the benefit of linear complexity, and a fixed overhead computation and memory cost for the window, as the size of the cache can only grow to a fixed, predetermined amount. However, this has the downside of constraining the information available in the cache, which may be needed for later predictions in the sequence. To illustrate, imagine streaming generation of an entire book with an LLM. If the cache can only contain tokens representing the average length of one chapter, then important information from previous chapters may be forgotten which could prove to be crucial for later steps of generation.

**Linear Prefill.** A crucial limitation of Streaming LLM (Xiao et al., 2023), H2O (Zhang et al., 2024b), and SnapKV (Li et al., 2024) lies in the handling of the prompt/prefill stage of the LLM. Each of the aforementioned works only considers that either 1) tokens are processed one-by-one during the prompt or 2) the prompt is processed with full quadratic attention and then proceeds to apply the relevant caching strategy during the generation phase. Being limited in this way, even a linear model like Streaming LLM exhibits

---

**Algorithm 1** Strided Prefill

---

**Require:** inputs, cache, model, stride_size
  **for** chunk **in** stride(inputs, stride_size) **do**
    **for** layer **in** model **do**
      KV ← cache.get()
      output, scores ← layer(chunk, KV)
      cache.update(chunk, scores)
    **end for**
  **end for**

---

slower latency than a quadratic prompt utilizing flash attention in the non-asymptotic regime (see Figure 6b). To bridge this gap, our method utilizes an attention kernel which processes fixed-sized chunks (strides) of the prompt in a single operation before adding the keys and values to our cascading cache. Specifically, for a sequence length of $S$, quadratic attention can be seen as having a stride size of $S$, Streaming LLM can be seen as having a stride size of 1, and our prefill method can be seen as having a stride size $K \in [1, S]$ that is somewhere in between. Algorithm 1 contains pseudocode for our strided prefill process, and Figure 5 contains an illustration. Our strided prefill allows for a more than two order of magnitude decrease in latency compared to single token processing and reduces latency by a factor of $6.8$ over quadratic flash attention when processing 1M tokens. This significant improvement delivers the benefits of linear attention on realistic, non-asymptotic sequence lengths.

## 3.1 Cascading KV Cache

Our cache is a generalization of sliding window attention which allows for important historical tokens to be kept in the cache history for a longer period of time. To accomplish this, we view a fixed sized sliding window KV cache $C$ with cardinality $|C|$ as a collection of sub-caches $C_i$ for $i \in [1, \ldots, N]$, each with cardinality $|C_i| \leq |C|$ such that $|\bigcup_{i=1}^{N} C_i| = |C|$. Assuming a sub-cache is full, each full sub-cache $\overline{C}_i$ evicts tokens when a new token is accepted into the sub-cache. However, each sub-cache accepts new tokens at a different rate, which in turn means that tokens may be discarded between the sub-caches and not solely at the end of the total cache window. An example of this process is depicted in Figure 4 (top), where the blue cache accepts all new incoming tokens. The green sub-cache, however, accepts only half of the tokens which are evicted from the blue sub-cache (every $2^{\text{nd}}$ iteration). The yellow sub-cache accepts half of the tokens evicted from the green sub-cache (every $4^{\text{th}}$ iteration) and so on.

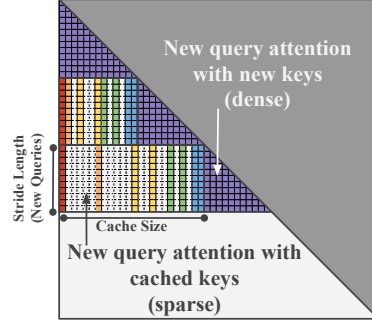

As Xiao et al. (2023) discovered, the initial attention sink tokens are vital for the stability of generation. Therefore, we too keep a fixed sub-cache of the initial tokens as attention sinks. With our method, a sink cache is a special case of a cascading cache, where the number of cascades (sub-caches) is set to 1, and it accepts all incoming tokens (*i.e.* using only the blue cache *and* red cache in Figure 4).

**Token Selection.** The process outlined in the preceding paragraph would result in a fixed heuristic pattern of tokens being dropped. However, such a heuristic pattern is not ideal, as it may be the case that the model naively holds onto tokens with limited value, while discarding important tokens. To remedy this, instead of discarding tokens naively, we dynamically select the most important tokens to retain by tracking the average attention score each token receives throughout time via an exponential moving average (EMA). We then selectively discard the token with the lower attention score EMA where possible. Given a hyperparameter $\gamma \in [0, 1]$, and a vector of attention scores for all keys in the cache $s_k^{(t)}$ at timestep $t$, we update the stored average attention scores $\boldsymbol{\mu}^{(t)}$ as, $\boldsymbol{\mu}^{(t+1)} = \gamma \boldsymbol{\mu}^{(t)} + (1 - \gamma) s_k^{(t)}$.

Figure 5: Our strided prefill. We first compute attention for a chunk (stride) of **new queries** and **new** + **cached** keys, forming a rectangular slice of the attention matrix at each step.

Consider the two sub-caches depicted in cases 2-4 of Figure 4. The blue sub-cache must accept all incoming tokens, while the green cache only accepts tokens every other iteration. Let the green cache be accepting tokens at the current timestep (case 2). When a new token comes in, it is added to the blue cache, which must then evict a token as the cache is full. The evicted token then goes to the green cache which accepts it unconditionally. The same process repeats on the next iteration, however, this time the green cache will not be accepting tokens (cases 3-4). At this step, when the blue cache accepts and subsequently evicts a token, we compare the attention score of the token evicted from the blue cache with the attention score of the newest token in the green cache. The token with the higher attention score is set (or remains) as the newest token in the green cache, while the token with the lower attention score is discarded. For a full pseudocode algorithm that covers cases 1-4 in Figure 4, see Algorithm 2 in the appendix.

**Positional Encoding.** We use the same positional encoding strategy as Streaming LLM, which reapplies positional encodings for each token in the cache according to its index within the cache. Assuming that tokens with a higher index are more recently added to the cache, the positional encoding function $\text{PE}(\mathbf{x}_i, i)$ applies positional encoding index $i$ to token $\mathbf{x}_i$. For example, if our cache holds the token indices from the original sequence $[0, 1, 3, 5, 7, 8]$, they would in turn receive positional encodings $[0, 1, 2, 3, 4, 5]$ which corresponds to their index within the cache.

**Circular Buffers.** Previous approaches have relied on tensor concatenation during cache add operations. However, this results in excessive copying operations, as each concatenation requires retrieving and storing all entries into a block of memory. We utilize more efficient circular buffers by tracking the location of the oldest token $\xi^{(t)}$ at timestep $t$ (where insertion should occur). We then increment $\xi$ after each insertion to the buffer such that $\xi^{(t+1)} = (\xi^{(t)} + 1) \mod |C_i|$. This way, at timestep $t$, we know that the oldest token in the buffer resides at $\xi^{(t)}$. When it is time to remove

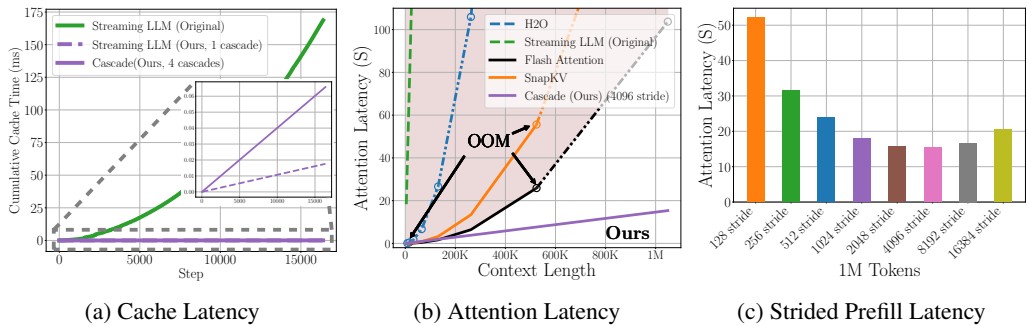

(a) Cache Latency          (b) Attention Latency          (c) Strided Prefill Latency

Figure 6: **Latency. a)** Our efficient cache implementation offers more than two orders of magnitude speedup over naive concatenation for cache add operations. **b)** Our strided prefill strategy is the only linear caching method which outperforms flash attention 2 latency on realistic sequence lengths (1M tokens). We fit a second degree polynomial (dotted lines) to predict latencies after quadratic models run out of memory on a 49GB GPU. **c)** Strided prefill latency for 1M tokens by different stride sizes.

a token, we simply overwrite the content of the memory address of $\xi^{(t)}$ rather than performing a costly concatenation.

**Cache Token Span.** Given previously outlined cascading cache, we have lengthened the span of the tokens which currently reside in the cache. The process outlined above effectively extends sliding window context length by allowing older tokens to remain as keys and values for a longer time, with gaps between tokens. Assuming that each sub-cache has the same capacity (*i.e.* $|\boldsymbol{C}| = |\bigcup_i^N \boldsymbol{C}_i| = N|\boldsymbol{C}_1|$), and each will accept tokens with a different frequency function defined as $\frac{1}{f(i)}$, then the approximate context length will be a summation over the cache sizes and the inverse of the frequency functions. For example, if each successive sub-cache accepts half of the tokens evicted from the previous cache, the total span of the cache $\tilde{S}$ becomes Equation (4). We can then calculate the overall sparsity of the cache as $1 - |\boldsymbol{C}|/S$ and sparsity of the cache window as $1 - |\boldsymbol{C}|/\tilde{S}$.

$$\tilde{S} = \sum_{i=1}^N f(i)|\boldsymbol{C}_i| = |\boldsymbol{C}_1| \sum_{i=1}^N 2^{i-1} = \frac{|\boldsymbol{C}|}{N} \sum_{i=1}^N 2^{i-1}. \tag{4}$$

## 4 EXPERIMENTS

**Setup.** We conduct experiments on streaming books (PG19) (Rae et al., 2019), Long Context Understanding (LongBench) (Bai et al., 2023b), book summarization (Booksum) (Kryściński et al., 2021), and 1M token passkey retrieval. We evaluate our method with pretrained transformers from the Llama3.1 (Dubey et al., 2024) and Qwen2 (Bai et al., 2023a) families of models. Please see Table 5 for model paths. In all our experiments, we keep the first 64 initial tokens as attention sinks. When considering the number of tokens in the cache, we always consider the sink tokens to be in addition to the cache size. Therefore a cache size of $|\boldsymbol{C}|$ has a total of $|\boldsymbol{C}| + 64$ total tokens. We set the EMA parameter $\gamma = 0.9999$. We use the cascade token acceptance policy depicted in Figure 4 and Equation (4), where each sub-cache accepts half of the tokens from the previous sub-cache. Unless otherwise indicated, our models use four cascading sub-caches. As Streaming LLM is a special case of our model, and 1 token per step is prohibitively slow, all results involving Streaming LLM results use our strided prefill strategy. Note that the strided prefill improves results over the original Streaming LLM with one token per step as shown in Figure 11. See Appendix B for further details.

### 4.1 LATENCY

We compare the latency of our cascading cache to the implementation from Xiao et al. (2023) which uses tensor concatenation to add/evict tokens from the cache. Our implementation utilizes circular buffers and the Triton compiler (Tillet et al., 2019) to create an efficient CUDA kernel for the caching operation. To perform this experiment, we initialize a cache with 64 sink tokens and a total window size of $|\boldsymbol{C}| = 16K$ with 4 and 1 cascades, which are equivalent to our Cascading KV Cache and Streaming LLM, respectively. We also initialize the original Streaming LLM that uses concatenation. We then process a total of 16K tokens into the cache, and report the cumulative time spent on caching operations. Our method with one cascade (equivalent to Streaming LLM) takes just $0.01\% = 1/10000$ of the total caching time of the original Streaming LLM, and our method

Table 1: PG19 Perplexity. Across all tested cache sizes, our cascading model maintains lower perplexity than baselines. Flash Attention 2 and Bigbird operate by stepping through the entire sequence with a stride equivalent to the total cache size. They perform attention at each step until reaching the end of the sequence. The Qwen model is excluded from 65K cache size due to having only 32K positional embeddings.

| Total Cache Size | Num. Books | Token Count | Model | Methods / Perplexity (↓) | | | | | | Cascading KV Cache (Ours) |
| --- | --- | --- | --- | --- | --- | --- | --- | --- | --- | --- |
| | | | | Flash Attn. 2 (strided) | | Big Bird (strided) | | Streaming LLM | | |
| 16384 | 91 | 9.78M | Qwen2₇B | 9.50 | (+0.36) | 10.65 | (+1.51) | 9.18 | (+0.04) | **9.14** |
| | | | LLaMA3.1₈B | 8.08 | (+0.36) | 12.76 | (+5.04) | 7.78 | (+0.06) | **7.72** |
| 32768 | 77 | 9.42M | Qwen2₇B | 9.26 | (+0.26) | 10.38 | (+1.35) | 9.05 | (+0.02) | **9.03** |
| | | | LLaMA3.1₈B | 7.86 | (+0.26) | 11.33 | (+3.73) | 7.65 | (+0.05) | **7.60** |
| 65536 | 62 | 8.25M | LLaMA3.1₈B | 7.73 | (+0.13) | OOM | (-) | 7.61 | (+0.01) | **7.60** |

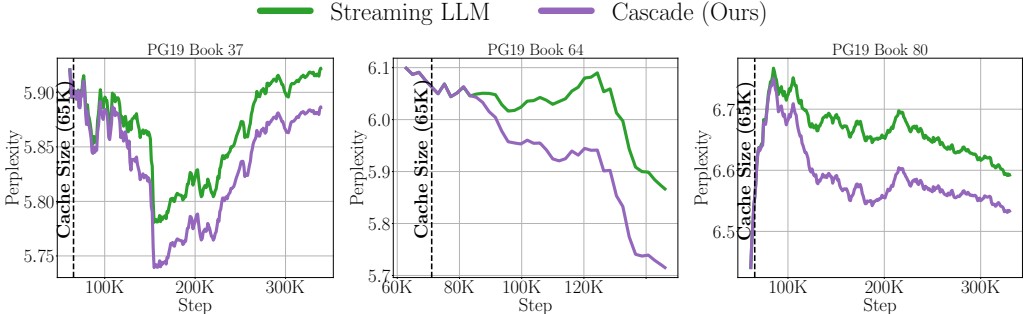

Figure 7: PG19. For Llama with a total cache size of 65K, our cascade model stays equivalent to Streaming LLM until the cache size is exceeded and our eviction policy and token selection begin. Our intelligent eviction policy leads to better perplexity for the total stream. Additional figures can be seen in Figures 14 and 15.

with 4 cascades takes just $0.038\% = 3.8/10000$ of the total caching time in Figure 6a. In Figure 6b, we show the overall attention latency, including caching, for a single attention layer processing 1M tokens. Our model uses a strided prefill of 4K with a cache size of 16K. Our method is the only method which processes 1M tokens faster than flash attention 2, and takes only $14.8\%$ of the time of quadratic flash attention. In other words, flash attention is 6.8 times slower than ours for processing 1M tokens. We also study the effect of the size of our strided prefill on overall 1M token latency in Figure 6b, finding that a stride of 4K delivers the lowest latency.

## 4.2 PG19

We measure perplexity on the PG19 test set (full-length books). Each book is streamed independently from start to finish without concatenation. We compare against a quadratic flash attention model, as well as BigBird and Streaming LLM, which are also training-free inference adaptations. We use three cache sizes of 16K, 32K, and 65K with a strided prefill of 1K. Since our cache is equivalent to Streaming LLM sequence lengths that are less than the total cache size, we only run each experiment on the subset of books which exceed the given cache size. Flash Attention 2 and Bigbird would exceed the GPU memory capacity, therefore, they are limited to processing books in chunk sizes equivalent to the total cache size. Results are displayed in Table 1. Our method delivers a consistently lower perplexity for all tested cache and model sizes. Additionally, we show examples of how our model behaves during the streaming process in Figure 7. After the cache size is exceeded, our eviction policy begins to differ from Streaming LLM, and our model tends to show lower perplexity due to the increased total token span in our cascading cache. Additional plots for Llama3.1 on all books exceeding 65K length can be seen in Figures 14 and 15.

## 4.3 BOOK SUMMARIZATION

We also evaluate our models on the booksum (Kryściński et al., 2021) book summarization dataset. For this experiment we compare against both linear and quadratic methods Table 2. We find that among linear methods, our models deliver a $4.48\%$ average performance increase over the best linear baseline rouge score (Lin & Hovy, 2003). We find that quadratic methods generally outperform our linear cache, although the performance increases come at a tremendous latency cost for longer input sequences as shown in Figure 6b.

Table 2: Booksum book summarization. Among linear baseline models, our Cascading KV cache offers a consistent improvement. Averaged over all models and metrics, ours performs 4.48% better than linear baselines.

| Method | Complexity | Llama 3.1$_{8B}$ | | | Qwen 2$_{7B}$ | | |
|---|---|---|---|---|---|---|---|
| | | Rouge-1 | Rouge-2 | Rouge-L | Rouge-1 | Rouge-2 | Rouge-L |
| Vanilla Transformer | $\mathcal{O}(N^2)$ | 35.37 | 7.29 | 21.24 | 31.70 | 5.70 | 17.8 |
| Snap KV | $\mathcal{O}(N^2)$ | 36.80 | 8.11 | 21.63 | - | - | - |
| H2O | $\mathcal{O}(N^2)$ | 35.62 | 7.42 | 21.16 | 31.10 | 5.47 | 17.58 |
| BigBird | $\mathcal{O}(N)$ | 33.36 | 6.55 | 18.59 | 20.83 | 2.31 | 12.62 |
| Streaming LLM | $\mathcal{O}(N)$ | 33.04 | 6.04 | 19.85 | 29.14 | 4.51 | 16.91 |
| Cascade (Ours) | $\mathcal{O}(N)$ | **34.47** | **6.63** | **20.52** | **30.34** | **5.02** | **17.54** |

(a) Ours 32K Total Cache Size

(b) Streaming LLM 32K Total Cache Size

(c) Ours 65K Total Cache Size

(d) Streaming LLM 65K Total Cache Size

Figure 8: Passkey Retrieval. For a total cache size of 65K, our Cascading KV Cache is able to maintain better than random (10%) accuracy even after 4 doublings of the context length beyond the cache size.

## 4.4 PASSKEY RETRIEVAL

For passkey retrieval, we evaluate Streaming LLM and our Cascading KV Cache, by generating a random 5 digit passkey which is hidden in a random point in the total sequence length. The rest of the text consists of random English words from the dictionary. We perform 20 trials for 5 insertion location ranges and 6 sequence lengths for a total of 600 retrievals. We calculate accuracy for each digit, counting a correct digit prediction if it falls in the proper place in the output sequence. Therefore, a model which outputs random digits would receive an accuracy of 10%. We evaluate total cache sizes of 32K and 65K and sequence lengths with 8 cascades and a strided prefill of 4K. Context lengths start from 32K and double until we reach 1M tokens. Results are shown in Figure 8. For both 32K and 65K cache sizes, Streaming LLM begins to show near random accuracy after the first doubling which exceeds the cache size, while our Cascading KV Cache is still better than random after 4 doublings of the context length.

## 4.5 LONGBENCH

We evaluate our method against other linear scaling models on the same subset of tasks as (Xiao et al., 2023) in the LongBench long context understanding benchmark (Bai et al., 2023b). We limit the total cache size of each model to be approximately 1/4 of the original prompt length $L$ of each input using the function $L' = 2^{\lfloor \log_2(L/4) \rfloor}$ and use a strided prefill of 512. Flash attention models truncate text from the middle of the input to achieve the desired cache length. The results are displayed in Figure 9. Averaged over all datasets and tested models, our cascading cache improves performance over the next best model by 12.13%. For tabular results, please see Table 9.

## 4.6 VISUALIZATION

To visualize the effect of our cache on the attention matrices, we reconstruct the full attention matrices of both Streaming LLM and our Cascading KV Cache using Llama3.1 8B on the first 8K tokens of the first book of the PG19 test set. We use a total cache size of 2048 and 4 cascades with a strided prefill of 256. The attention matrices are displayed in Figure 10. Naive sliding window attention Figure 10 (a,c) forms short static barrier where tokens are evicted regardless of their importance. Our cache Figure 10 (b,d) retains tokens in history for a longer time where have influence over future predictions, effectively increasing the available knowledge.

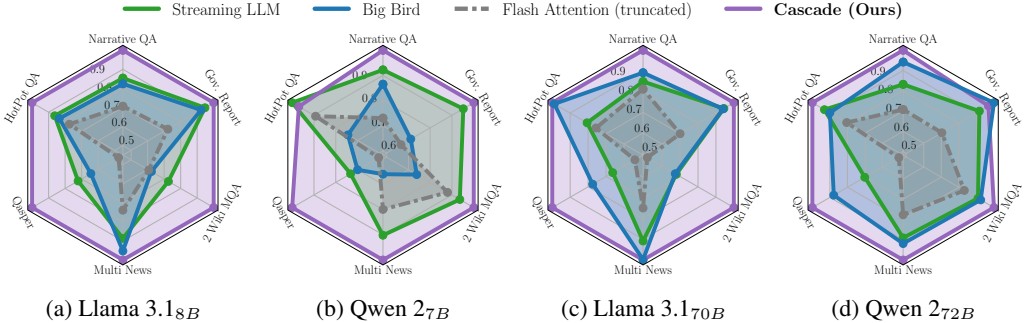

Figure 9: LongBench. Our cascade model consistently outperforms linear inference baselines. All models are limited to a context window which is roughly 1/4 of the total original prompt length. Averaged over all models and datasets, our Cascading KV cache results in an average performance gain of 12.3%. Please see Table 9 for a tabular presentation of results.

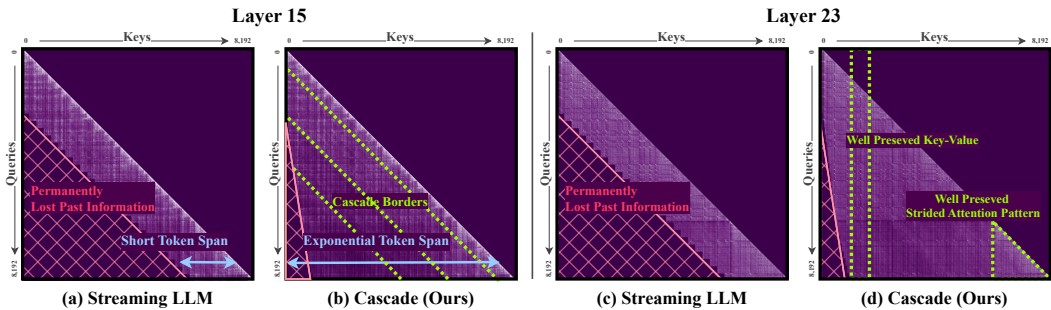

Figure 10: Attention matrix reconstruction for Streaming LLM's sink cache and our Cascading Cache. Both methods result in $\mathcal{O}(n)$ inference time complexity with the same total cache size ($|C| = 2048$).

We demonstrate the cascade boundaries of our proposed KV eviction policy in Figure 10(b), where the sparsity of each cascade gradually increases with the size of the cascade. Additionally, our method preserves the attention patterns more thoroughly than Streaming LLM, such as the annotated preserved key value in Figure 10(d) which falls well outside of the range of the sliding window pattern. For more attention visualizations, please see Figure 17 in the appendix.

Table 3: The token selection process outlined in Section 3 is crucial for creating dynamic attention patterns.

| KV Cache ($|C| = 2048$) | LLaMA3.1$_{8B}$ PPL ↓ |
|---|---|
| Streaming LLM | 8.03 |
| Ours w/o token selection | 8.03 |
| Ours w/ token selection | **7.88** |

## 4.7 ABLATION STUDY

To study the effect of different parts of our model, we provide three ablation studies including the effect of the strided prefill and token selection using the first book of PG19, and the effect of sparsity induced by the number of cascades. The effect of the strided prefill is shown in Figure 11. We find a decrease in perplexity with an increasing stride size. Intuitively, this comes from the fact that a larger stride provides a larger dense window at the leading edge of the attention matrix as shown in Figure 5. We study the token selection process outlined in Section 3 in Table 3 and find that without the token selection process, our model matches the performance of Streaming LLM, which highlights the importance of selecting higher scoring tokens. Lastly, we study the effect of the number of cascades, and thus overall sparsity, in Figure 12. For this experiment, we use a total cache size of 4K and consider context

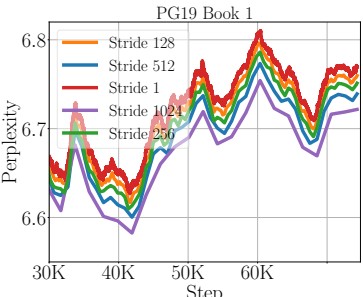

Figure 11: Streaming LLM with our strided prefill achieved a progressively better perplexity and latency (Figure 6c) when increasing the stride due to a larger region of dense attention.

lengths from 4K which double until 262K. We average the passkey retrieval accuracy over all insertion locations. We find that accuracy steadily increases until the number of cascades exceeds 8 (more than 96% window sparsity calculated by $1 - |C|/\tilde{S}$).

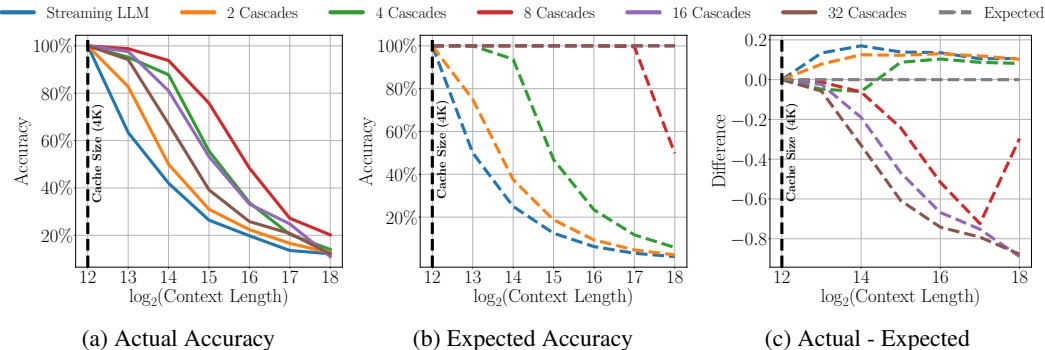

Figure 12: The effect of more cascades. **a)** For a fixed cache size, increasing the number of cascades leads to more sparsity. We find passkey retrieval accuracy increases until 8 total cascades. **b)** Expected accuracy using total token span (Equation (4)) as a rough predictor. **c)** measuring the difference between predicted and actual accuracy (Figure 12a - Figure 12b), we see that token span remains a strong predictor until the number of cascades exceeds four.

Interestingly, the token span in the cache remains a good predictor of accuracy for a moderate number of cascades. Given a token span, we may roughly calculate the expected accuracy in Figure 12b by considering the probability that the passkey falls within the span of the tokens. For example, with a token span of $\tilde{S} = 1024$, and a context size of $2048$, we would expect an accuracy of approximately $50\%$. We find the token span to be a reliable predictor of accuracy until 4 cascades (Figure 12c).

## 5    LIMITATIONS & FUTURE WORK

As described above Equation (4) and visualized in Figure 12, the overall sparsity of our model increases as the number of cascades $N$ increases. This introduces a trade-off, as demonstrated in Figure 12a, where performance improves up to a point but decreases once the number of cascades exceeds eight. Therefore, while our method enables substantial context length extrapolation, this process is not unbounded. Eventually, tokens must be discarded to maintain linear inference complexity, which inherently limits the scope of extrapolation. A promising direction for future work involves removing the need to discard any tokens while preserving sub-quadratic complexity. One potential solution could involve developing methods for logarithmic complexity searches for query-key similarity within the KV cache. By efficiently identifying the top-k relevant key-value pairs, such an approach could eliminate the need for token eviction and allow the model to maintain an overall complexity of $\mathcal{O}(N \log N)$. This would open new avenues for expanding the context memory of transformer models without compromising efficiency.

## 6    CONCLUSION

In this paper, we introduced a novel, training-free method for extending the context memory of streaming LLMs, offering significant improvements without increasing computational complexity. Our approach treats the fixed-size KV cache as a series of cascading sub-caches, allowing for dynamic token retention based on their historical importance. By selectively preserving high-impact tokens and evicting less critical ones, our method effectively extends the context window far beyond the limitations of traditional sliding windows. Our results demonstrate clear performance gains: a 12.13% average improvement in LongBench, a 4.48% boost in book summarization tasks, and superior passkey retrieval accuracy at 1M tokens, maintaining a significant edge even after four doublings of the context size. Additionally, our linear prefill strategy eliminates the quadratic complexity of previous approaches, achieving latency reduction by a factor of 6.8 compared to flash attention 2. These advancements highlight the potential of our method to significantly improve the efficiency of LLMs, making it a practical and impactful solution for both research and real-world applications that require long-context processing.

REPRODUCIBILITY STATEMENT

In order to aid in reproducibility of our experiments, we provide our code which is available online[2]. We also provide exact pretrained model URL's which are listed in Table 5. We provide an algorithm for our strided prefill method in Algorithm 1 and a complete algorithm for our Cascading KV Cache in Algorithm 2 and modified flash attention kernnel in Algorithm 3. We have explained the parameters and computation budgets for all experiments in Section 4. As our method is deterministic and does not require a stochastic training or generation process, we have omitted error bars in our results.

ACKNOWLEDGMENTS

This work was supported by Institute for Information & communications Technology Planning & Evaluation(IITP) grant funded by the Korea government(MSIT) (No. RS-2019-II190075, Artificial Intelligence Graduate School Program(KAIST)); (No. RS-2019-II191906, Artificial Intelligence Graduate School Program(POSTECH)); (No. RS-2024-00459797, Development of ML compiler framework for on-device AI); (No. RS-2022-II220713, Meta-learning Applicable to Real-world Problems); and (No. RS-2022-00187238, Development of Large Korean Language Model Technology for Efficient Pre-training).

This work was supported by the National Research Foundation of Korea(NRF) grant funded by the Korea government(MSIT) (No. RS-2023-00256259); and (No. RS-2024-00354947).

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

## A  APPENDIX

## B  EXPERIMENTAL SETTINGS

For experiments utilizing 8B model sizes, we use one NVIDIA A6000 (49GB) GPU, and for experiments utilizing 70B model sizes, we utilize 4 NVIDIA A100 GPUs. Quadratic models which use Flash Attention 2 (Dao, 2023) utilize the official cuda kernel, while our method utilizes a modified triton (Tillet et al., 2019) Flash Attention 2 kernel.

## C  HEAD POLICY AND HEAD REDUCTION.

When making a decision for token selection, we may apply the same homogeneous decision across all heads. Likewise, we may allow the heads to behave independently as illustrated in Figure 13a. Additionally, as models may make use of Grouped Query Attention (GQA) (Ainslie et al., 2023), the number of attention heads may differ between queries and keys. Therefore, for both cases of homogeneous and independent heads, we need to select a head reduction function which will reduce the head dimension in the attention matrix to 1 (homogeneous heads) or K (key-value heads in GQA). We perform an ablation on the PG19 dataset to explore different options of head reduction functions and head policies in Figure 13b. We find that in all cases, independent heads outperform homogeneous heads. Among the independent heads, we find that mean and max reductions resulted in similar performance, while a median reduction resulted in slightly worse performance. We utilize independent heads and max pooling over attention scores in our experiments.

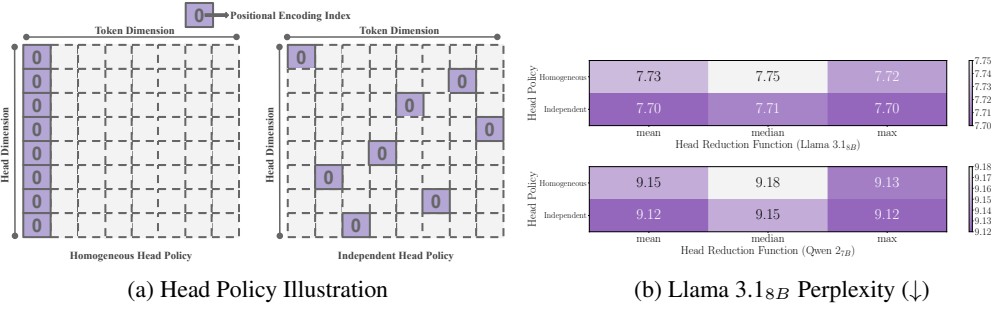

(a) Head Policy Illustration                              (b) Llama 3.1$_{8B}$ Perplexity ($\downarrow$)

Figure 13: Head Reduction Function and Head Policy Ablation

| Model | Huggingface Path | Experiment |
|-------|------------------|------------|
| LLaMA3.1$_{8B}$ (Dubey et al., 2024) | meta-llama/Meta-Llama-3-8B | PG19 |
| LLaMA3.1$_{8B}$ Instruct (Dubey et al., 2024) | meta-llama/Meta-Llama-3-8B-Instruct | Booksum,Longbench,Ablation,Passkey |
| LLaMA3.1$_{70B}$ Instruct (Dubey et al., 2024) | meta-llama/Meta-Llama-3-70B-Instruct | Longbench |
| Qwen2$_{7B}$ (Bai et al., 2023a) | Qwen/Qwen2-7B | PG19 |
| Qwen2$_{7B}$ Instruct (Bai et al., 2023a) | Qwen/Qwen2-7B-Instruct | Booksum,Longbench |
| Qwen2$_{72B}$ Instruct (Bai et al., 2023a) | Qwen/Qwen2-72B-Instruct | LongBench |

Table 5: Huggingface model paths used in our experiments.

Table 4: MMLU. We find that our Cascading KV Cache outperforms all linear models overall.

| Model | Method | Complexity | Humanities | Soc. Science | Other | STEM | Overall |
|-------|--------|-----------|-----------|--------------|-------|------|---------|
| Llama 3.1$_{8B}$ | Flash Attention 2(Full Context) | $\mathcal{O}(N^2)$ | 61.59 | 76.47 | 73.09 | 56.01 | 67.00 |
| | Vanilla (Truncated) | $\mathcal{O}(N)$ | 61.23 | 76.08 | 73.06 | 55.79 | 66.80 |
| | Streaming LLM | $\mathcal{O}(N)$ | 61.57 | 76.37 | 73.19 | 55.92 | 66.98 |
| | Cascade (Ours) | $\mathcal{O}(N)$ | 61.45 | 76.63 | 73.25 | 56.23 | **67.11** |
| Qwen 2$_{7B}$ | Flash Attention 2 (Full Context) | $\mathcal{O}(N^2)$ | 63.12 | 80.40 | 74.64 | 64.07 | 70.99 |
| | Vanilla (Truncated) | $\mathcal{O}(N)$ | 62.61 | 80.21 | 74.64 | 64.12 | 70.81 |
| | Streaming LLM | $\mathcal{O}(N)$ | 63.04 | 80.47 | 74.61 | 63.84 | 70.86 |
| | Cascade (Ours) | $\mathcal{O}(N)$ | 62.93 | 80.40 | 74.57 | 63.88 | **70.87** |

---

**Algorithm 2** Cascading Sink Cache Algorithm (repeat for keys and values)

---

**Require:** cascade_cache_buf_array, sink_cache_buf, item_to_cache
  **if** not sink_cache_buffer.is_full() **then**
    sink_cache_buffer ∪ item_to_cache
    return
  **end if**
  **for** cache_buf **in** cascade_cache_buf_array **do**
    **if** cache_buf.is_accepting_tokens() **then**
      **if** not cache_buf.is_full() **then**        ▷ add item to cache which is not full
        cache_buf ∪ item_to_cache
        return
      **else**        ▷ evict an item from the cache and continue
        cache_buf ∪ item_to_cache
        item_to_cache ← cache_buf.evict_oldest()      ▷ reset variable for next iteration
      **end if**
    **else**
      **if** not cache_buf.is_full() **then**    ▷ eager add to unfilled cache to avoid naively evicting
        cache_buf ∪ item_to_cache
        return
      **else**        ▷ token selection (newest in cache vs. incoming token)
        newest_item ← cache_buf.get_newest_item()
        newest_score ← cache_buf.get_score(newest_item)
        item_score ← cache_buf.get_score(item_to_cache)
        **if** item_score > newest_score **then**
          cache_buf.evict_newest()
          cache_buf ∪ item_to_cache
        **end if**
        return
      **end if**
    **end if**
    update_positional_encoding_indices()
  **end for**

---

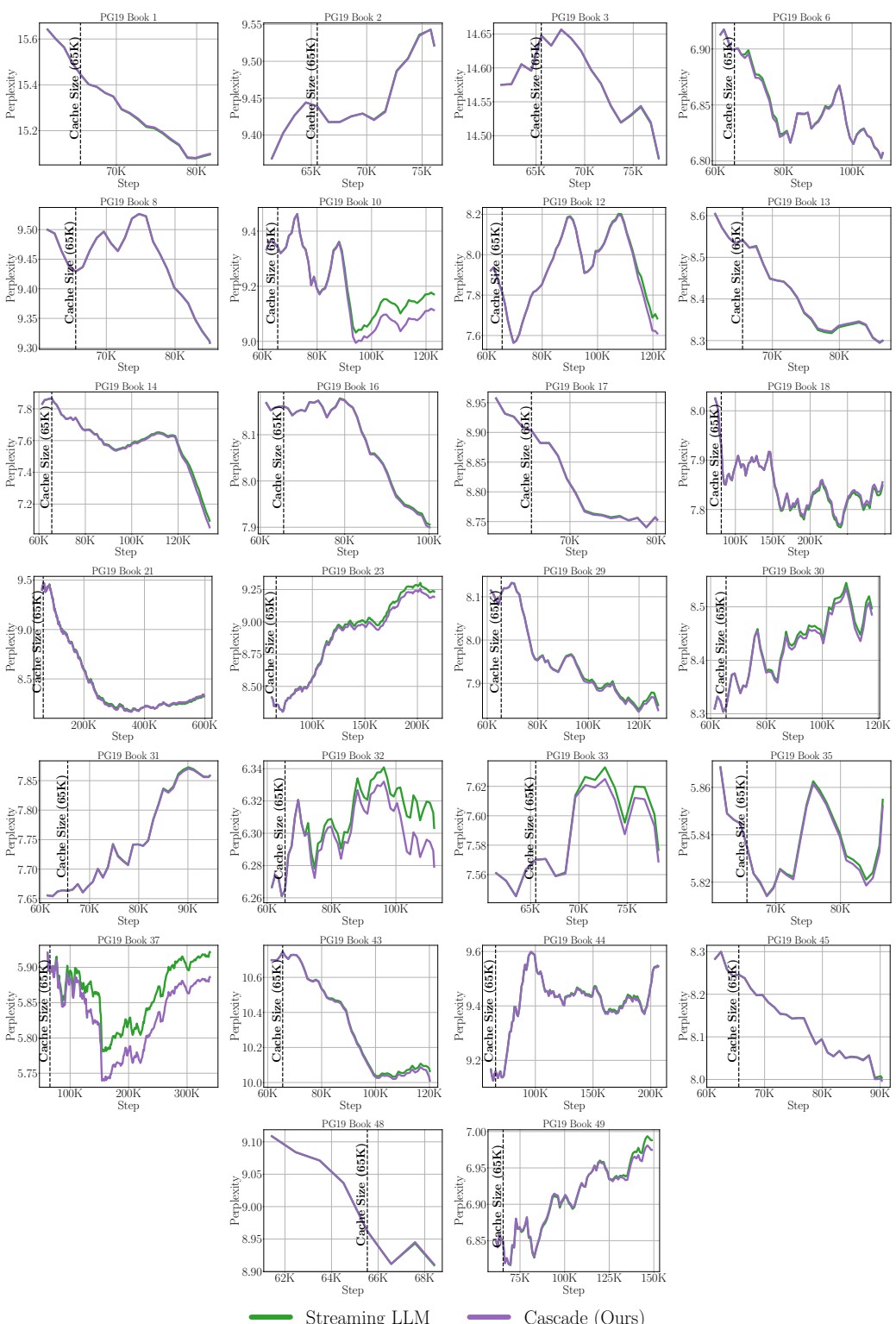

Figure 14: Additional plots for all of the books in the subset of PG19 books which exceeds 65K in length. The model used is Llama 3.1 8B. This chart is a complement to Figure 7

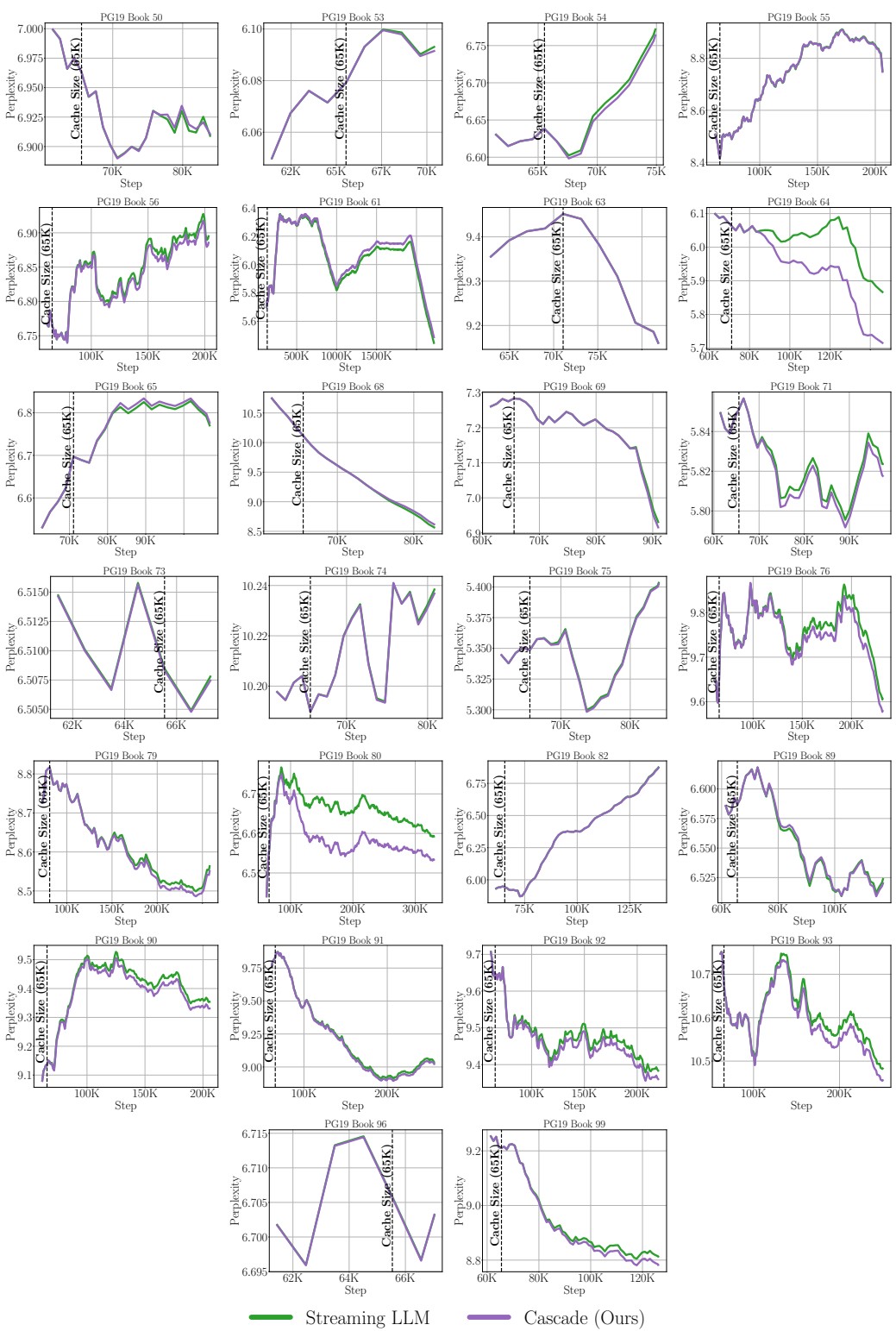

Figure 15: Additional plots for all of the books in the subset of PG19 books which exceeds 65K in length. The model used is Llama 3.1 8B. This chart is a complement to Figure 7

---

**Algorithm 3** Token Selection EMA Accumulation (in the context of Flash Attention 2 kernel)

---

**Require:** score_cache, queries, keys, m, EMA beta

  **for** i **in** chunk(queries) **do**

    Load $Q_i$ from HBM to on-chip SRAM from queries

    **for** j **in** chunk(keys) **do**

      Load $K_j, V_j$ from HBM to on-chip SRAM from keys

      On Chip, Compute $S_{ij} = Q_i K_j^\top$

      On Chip, update $m_i$ (update rolling max a la flash attention 2)

      On Chip, update $l_i$ (update normalization constant a la flash attention 2)

      On Chip, Compute $\hat{S}_{ij}$ (max-adjusted exponential before softmax normalization)

      On Chip, update $O_i$ (update output vector a la flash attention 2)

      On Chip, calculate EMA coeff. $C_{\text{EMA}}$ for $Q_i$,

          $C_{\text{EMA}} = \beta^k(1 - \beta) \forall k \in [\text{len(queries)} - (\text{q\_chunk\_indices} + 1)]$

      On Chip, calculate inner loop steps completed $\gamma$ and remaining $\rho$

      Write, Atomic Sum to score_cache += col_sum$((\hat{S}_{ij}/(l_i + l_i * \frac{\rho}{\gamma})) * C_{\text{EMA}})$

    **end for**

  **end for**

---

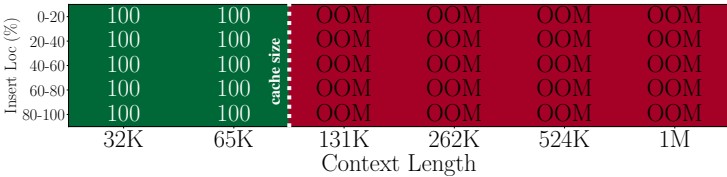

Figure 16: Passkey results for vanilla Llama3.1 as measured on an NVIDIA A6000 GPU with 49GM memory and a cache size of 65K. According to the llama whitepaper (Dubey et al., 2024), it was finetuned for 131K positional embeddings until it achieved 100% accuracy on a passkey retrieval task. Therefore, given a big enough cache, it achieves perfect accuracy up to the cache size (or 131K). Our model however, can extend high accuracy numbers well beyond the cache size (see Figure 8).

| Model | 16K | 32K | 65K | 131K | 262K | 524K | 1M |
|---|---|---|---|---|---|---|---|
| Minference | 5.11 | 12.84 | 28.41 | 60.47 | OOM | OOM | OOM |
| Cascading KV Cache | 9.58 | 19.71 | 40.11 | 80.84 | 164.47 | 334.17 | 665.33 |

Table 6: Latency (S) as compared to Minference for 1M tokens. This table shows latency throughout all layers, which differs from that shown in Figure 6b which shows attention latency for a single layer. Minference goes OOM after 131K on a 49GB GPU due to requiring all tokens in cache for the forward pass. Our model uses a cache size of 16K with a stride of 4K, which are the same settings used in Figure 6b.

## C.1 ADDITIONAL LONGBENCH/INFINITEBENCH BASELINES

In Table 9, we show tabular results for the data displayed in Figure 9. We also add three additional models (MInference (Jiang et al., 2024), PyramidKV (Cai et al., 2024) and Cascade + MInference). The settings between these extra methods contrasts with our setting in various ways (Table 7), as Minference supports a sparse prefill with dense decode and PyramidKV supports a dense prefill with a sparse decode. As our method is the only one which supports both sparse prefill and decode, we truncate the middle of the prompt for MInference and PyramidKV in the same way we do for flash attention in Figure 9.

Table 7: Prefill and decoding sparsity for additional models.

| Method | Sparse Prefill | Sparse Decode |
|---|---|---|
| MInference | ✓ | ✗ |
| PyramidKV | ✗ | ✓ |
| Cascade (OURS) | ✓ | ✓ |

In addition to this, we have also provded an example of our cache integrated with the official MInference kernel. This allows for using our our linear strided prefill and cascading cache in conjunction with MInference. The key difference in MInference+Cascade is that the leading chunk (stride) of the attention matrix uses MInference instead of dense attention. As MInference is very sparse, this allows for greatly increasing the size of the stride without adding much computation overhead while computing each row of the attention matrix. For this experiment MInference+Cascade used a stride of 100K.

Infinite bench (Zhang et al., 2024a). results can be seen in Table 10 using the same truncation strategy for MInference as outlined above.

## C.2 ADDITIONAL LATENCY EXPERIMENT

In Table 8, we evaluate two additional models (MInference and MInference + Cascade) in terms of latency for a single self attention layer. MInference shows a tendency to scale super-linearly, as each doubling of the context results in a more than doubling of the latency. Because of this, MInference + cascade begins to outperform the latency of plain MInference after sequence lengths of 131K. This decrease in latency is due to the fact that MInference + Cascade uses MInference as the leading chunk in the attention matrix (with a 100K stride) which bounds the total size of the attention matrix of a single attention head at each step to be $\in \mathbb{R}^{100K \times (100K + |C|)}$ where $|C|$ is the total cache size.

Table 8: Additional 1M token latency (S) results. While our cascade model with a stride of 4096 maintains the best overall latency, MInference + cascade also shows linear scaling in addition to added performance benefits shown in Table 9. MInference + cascade begins to outperform plain MInference in terms of latency on sequences over 131K in length. For further discussion of these results, please see Appendix C.2.

| Model | 16K | 32K | 65K | 131K | 262K | 524K | 1M |
|---|---|---|---|---|---|---|---|
| MInference | 0.26 | **0.28** | **0.78** | 2.26 | 6.39 | 16.84 | OOM |
| MInference + Cascade | 0.34 | 0.68 | 1.3 | 3.04 | 5.77 | 11.88 | 24.13 |
| Cascade | **0.22** | 0.46 | 0.94 | **1.93** | **3.81** | **7.66** | **15.34** |

Table 9: Tabular display of LongBench radar plot results from Figure 9. Higher scores are better (↑). Total cache sizes are based on the length of the original prompt $L$. For Llama 3.1 8B Instruct we added three additional models (MInference, PyramidKV and Cascade + MInference). For more details regarding these models, please refer to Appendix C.1.

| Total Cache Size | Model | Cache | Narrative QA | HotPot QA | Qasper | Multi News | 2 Wiki MQA | Gov. Report | Mean |
|---|---|---|---|---|---|---|---|---|---|
| $2^{\lfloor \log_2(L/4) \rfloor}$ | LLaMA3.1$_{8B}$ Instruct | Streaming LLM | 22.57 | 40.78 | 23.89 | 20.69 | 23.46 | 26.82 | 26.37 |
| | | Flash Attention 2 (truncated) | 18.67 | 36.59 | 15.77 | 17.24 | 19.62 | 20.46 | 21.39 |
| | | Big Bird | 21.78 | 39.46 | 21.33 | 22.23 | 20.13 | 26.15 | 25.18 |
| | | MInference (truncated) (Jiang et al., 2024) | 20.90 | 39.79 | 19.77 | 23.19 | 21.52 | 29.15 | 25.72 |
| | | Pyramid KV (truncated) (Cai et al., 2024) | 20.99 | 39.79 | 19.86 | 22.20 | 21.77 | **29.20** | 25.63 |
| | | Cascade (**Ours**) | **26.43** | **47.26** | **33.12** | **23.33** | **32.33** | 28.32 | **31.8** |
| | | Cascade (**Ours**) + MInference (Jiang et al., 2024) | 29.84 | 55.22 | 36.51 | 23.14 | 48.85 | 28.2 | 36.96 |
| $2^{\lfloor \log_2(L/4) \rfloor}$ | Qwen2$_{7B}$ | Streaming LLM | 18.95 | **38.54** | 20.97 | 17.65 | 32.15 | 24.96 | 25.54 |
| | | Flash Attention 2 (truncated) | 15.01 | 34.4 | 17.28 | 15.64 | 30.21 | 17.47 | 21.67 |
| | | Big Bird | 17.78 | 28.68 | 20.05 | 12.9 | 25.36 | 18.64 | 20.57 |
| | | Cascade (**Ours**) | **20.55** | 37.36 | **28.65** | **19.57** | **34.35** | **26.22** | **27.78** |
| $2^{\lfloor \log_2(L/4) \rfloor}$ | LLaMA3.1$_{70B}$ | Streaming LLM | 25.72 | 42.39 | 24.5 | 20.93 | 31.78 | 27.67 | 28.83 |
| | | Flash Attention 2 (truncated) | 24.62 | 39.77 | 19.93 | 17.26 | 24.23 | 20.51 | 24.39 |
| | | Big Bird | 27.02 | 52.0 | 28.63 | 23.0 | 31.51 | 27.53 | 31.61 |
| | | Cascade (**Ours**) | **30.3** | **52.61** | **36.97** | **23.04** | **46.82** | **29.3** | **36.51** |
| $2^{\lfloor \log_2(L/4) \rfloor}$ | Qwen2$_{72B}$ | Streaming LLM | 20.8 | 50.08 | 20.68 | 17.97 | 43.83 | 27.83 | 30.2 |
| | | Flash Attention 2 (truncated) | 17.69 | 43.25 | 14.96 | 15.68 | 40.08 | 21.35 | 25.5 |
| | | Big Bird | 23.55 | 48.41 | 25.89 | 18.54 | 44.58 | **30.35** | 31.89 |
| | | Cascade (**Ours**) | **25.0** | **53.78** | **29.48** | **20.17** | **48.22** | 29.4 | **34.34** |

Table 10: InfiniteBench (Zhang et al., 2024a) results.

| Total Cache Size | Model | Cache | en.MC | en.QA | en.Sum | Mean |
|---|---|---|---|---|---|---|
| 32768 | LLaMA3.1$_{8B}$ Instruct | Streaming LLM | 46.72 | 13.98 | 30.8 | 30.5 |
| | | Minference (truncated) | 46.72 | 14.96 | **32.25** | 31.31 |
| | | Cascade (**Ours**) | **56.77** | **17.69** | 31.50 | **35.32** |

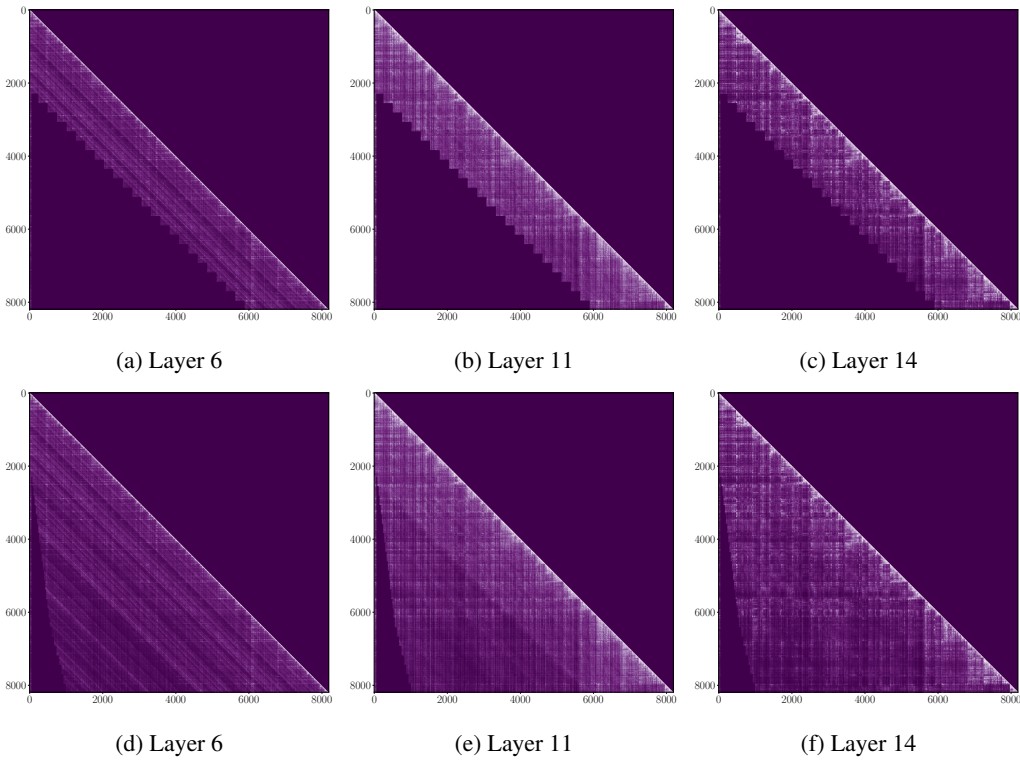

(a) Layer 6     (b) Layer 11     (c) Layer 14

(d) Layer 6     (e) Layer 11     (f) Layer 14

Figure 17: Attention matrix reconstructions for Streaming LLM Figures 17a to 17c and our Cascading KV Cache Figures 17d to 17f on first 8K tokens of the first book of (PG19).

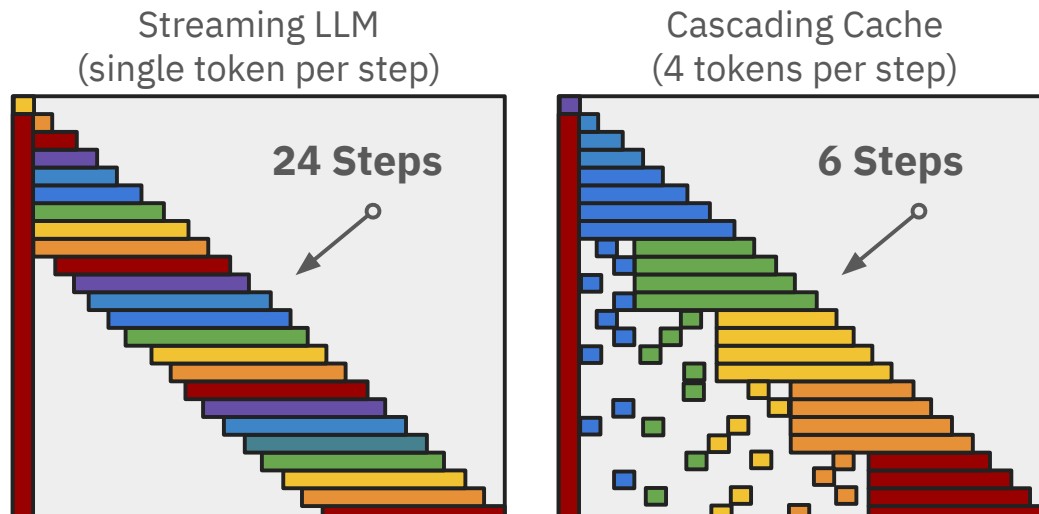

Figure 18: Illustration contrasting the prefill strategy of Streaming LLM vs our Cascading KV Cache. The original Streaming LLM does a complete forward pass for every row of the attention matrix which causes the poor latency of Streaming LLM in Figure 6b. Our method, however, can process a chunk of tokens during each forward pass of the prefill leading to a reduction in the number of forward passes necessary to process the entire prompt.

