# OpenReview forum: "Training Free Exponential Context Extension via Cascading KV Cache"
_ICLR.cc/2025/Conference — ICLR 2025 Poster_

### Official Review · Reviewer_4zw7 · 2024-11-01

**Soundness:** 3
**Presentation:** 2
**Contribution:** 2
**Rating:** 6
**Confidence:** 4

**Summary:**

In order to enable LLMs to preserve a long context without increasing the KV cache size, this paper proposes a training-free KV cache eviction policy with a cascading sub-cache algorithm. The algorithm allows one sub-cache window to accept a subset of tokens evicted from the previous sub-cache. This paper also introduces a linear prefill strategy that avoids the restrictive quadratic prompt complexity during the prefill stage by processing fixed-sized chunks (strides) of the prompt.

**Strengths:**

1. The experiments and conclusions are sufficient, including the evaluation of latency and accuracy on different tasks and LLMs. Also, the display of the experimental figures are clear.
2. This work builds a new paradigm of KV cache with cascading sub-cache and effectively extends the context length without increasing complexity or requiring additional training. This work provides a promising solution for efficient edge-side LLM inference.

**Weaknesses:**

1. I understand the proposed method is a generalization (more flexible version) of sliding window attention, but the main problem is that this paper does not clearly explain why the proposed cascading sub-cache is effective for KV cache eviction (intuitively). It is better to explain more about the background of cascading structure and the key motivations/observations/insights for introducing cascading KV cache.
2. Although experiments are sufficient, the baselines are kind of weak. For example, when evaluating on LongBench, there are more recent dynamic KV cache method such as PyramidKV [1], PyramidInfer[2], InfLLM [3], Quest [4], and FastGen [5]. It will be better to compare with one/two more recent dynamic KV cache method to further support your conclusions.
3. In Section 3.1, it is not clear that why use trunks can achieve linear attention complexity during the prefill stage, and what is the key difference between the proposed method and FlashAttention which also uses tiling to save the memory.
4. From Section 3.2, each sub-cache relies on the previous sub-cache. The sequential procedure seems to have an impact on the inference efficiency. How to avoid this problem and achieve much higer efficiency compared with StreamingLLM.
5. Token selection algorithm leverages exponential moving average (EMA) to track the historical attention score. This method seems similar to directly using accumulated attention score proposed by H2O. Can I understand this process as considering the different importance degrees of current and historical attention score based on the method of H2O, which is controlled by the hyper-parameter gamma.

Related references:
[1] PyramidKV: Dynamic KV Cache Compression based on Pyramidal Information Funneling (arXiv 2024)
[2] PyramidInfer: Pyramid KV Cache Compression for High-throughput LLM Inference (ACL 2024)
[3] InfLLM: Training-Free Long-Context Extrapolation for LLMs with an Efficient Context Memory (arXiv 2024)
[4] Quest: Query-Aware Sparsity for Efficient Long-Context LLM Inference (ICML 2024)
[5] Model Tells You What to Discard: Adaptive KV Cache Compression for LLMs (ICLR 2024)

**Questions:**

Please refer to the weakness part for questions.

---

> ### Author Response · Authors · 2024-11-19
> **Author Response I**
>
> Thank you for your time in reviewing our work and highlighting the strengths of our method in increasing the context length while lowering the complexity without training. Please see our responses to your additional comments below:
>
> ---
>
> **[W1]** [...] paper does not clearly explain why the proposed cascading sub-cache is effective for KV cache eviction (intuitively). It is better to explain more about [...] background [...] motivations/observations/insights for introducing cascading KV cache.
>
> **[Response]** We apologize if this was not adequately highlighted in the original submission. We have revised the text of the intro to include more detailed motivation. To summarize, figure 1 visualizes the forgotten context tokens which are a side effect of sliding windows. Figure 2 shows the effect of this forgotten context on the passkey retrieval task, which suffers almost immediately as the context exceeds the sliding window. The motivation for the cascade pattern was derived as a simple and effective way to allow multiple exit paths for tokens such that unimportant tokens may be *conditionally* evicted before reaching the end of the cache buffer.
>
> ---
>
> **[W2]** Although experiments are sufficient, the baselines are kind of weak. For example, when evaluating on LongBench, there are more recent dynamic KV cache method such as PyramidKV [1], PyramidInfer[2], InfLLM [3], Quest [4], and FastGen [5]. It will be better to compare with one/two more recent dynamic KV cache method to further support your conclusions.
>
> **[Response]**
>
> Based on your feedback as well as other reviewers, we have added two new baselines,  Minference and Pyramid KV, on LongBench. As Minference relies on fully dense decoding and Pyramid KV requires quadratic prompts, we truncate the input according to the standard in LongBench evaluation. All results below utilize Llama 3.1 8B Instruct.
>
> For LongBench, we find that our method outperforms both Minference and PyramidKV on all but the Gov. Report Dataset which shows a thin margin between all models as seen in Figure 9. The average performance increase over all datasets is still heavily in favor of our cascading cache.
>
> |  | Narrative QA | HotPotQA | Qasper | Multi News | 2WikiMQA | Gov. Report | Avg. |
> |---|---|---|---|---|---|---|---|
> | Pyramid KV | 20.99 | 39.79 | 19.86 | 23.2 | 21.77 | **29.2** | 25.63 |
> | Minference | 20.9 | 39.79 | 19.77 | 23.19 | 21.52 | 29.15 | 25.72 |
> | Streaming LLM | 22.57 | 40.78 | 23.89 | 20.69 | 23.46 | 26.82 | 26.37 |
> | Cascade (Ours) | **26.43** | **47.26** | **33.12** | **23.33** | **32.33** | 28.32 | **31.8** |
>
> For InfiniteBench we allow a total cache size of 32K. we observe the same pattern as LongBench, whereby our model outperforms Minference by a large margin (21.5% and 18.2%) on the first two datasets, and only underperforms Minference by 2.3% on the last dataset where the overall differences between models is much smaller.
>
> |  | en.MC | en.QA | en.Sum | Avg. |
> |-|-|-|-|-|
> | Streaming LLM | 46.72 | 13.98 | 30.8 | 30.5 |
> | Minference | 46.72 | 14.96 | **32.25** | 31.31 |
> | Cascade (Ours) | **56.77** | **17.69** | 31.5 | **35.32**|
>
> ---
>
> Continued in next comment...

---

> ### Author Response · Authors · 2024-11-19
> **Author Response II**
>
> **[W3]** In Section 3.1, it is not clear that why use trunks can achieve linear attention complexity during the prefill stage, and what is the key difference between the proposed method and FlashAttention which also uses tiling to save the memory.
>
> **[Response]** We are unsure what is meant by "use trunks." We assume this question relates to how we can achieve linear complexity during the prefill stage compared to Streaming LLM. There are actually two models in figure 6b which achieve linear complexity during prefill (Streaming LLM, and ours). The main difference is that the single token per step prefill of Streaming LLM is slow enough that it cannot beat quadratic attention implementations on realistic context lengths. Asymptotically, if the x-axis of figure 6b were extended far enough, Streaming LLM would eventually show lower latency than quadratic attention. Our method becomes faster that streaming LLM due to the fact that in one forward pass of the prefill stage, we process a chunk of tokens as opposed to processing a single token per forward pass as done in the original Streaming LLM prefill. We have added an illustration of this difference to the appendix in Figure 18. This change allows for a drastic speedup, improved performance (Figure 11), and maintains a fixed memory overhead when processing any arbitrary context length.
>
> The key difference between our method and flash attention is that flash attention still has quadratic latency ($\mathcal{O}(n^2)$) while ours has a fixed cache size which caps the second dimension at a constant $k$, achieving an overall complexity of ($\mathcal{O}(nk)$), which can be expressed as $\mathcal{O}(n)$ after dropping constants by convention.
>
> The two aforementioned aspects combined are what allows our method to show superior latency in figure 6b.
>
> ---
>
> **[W4]** From Section 3.2, each sub-cache relies on the previous sub-cache. [...] seems to have an impact on the inference efficiency. How to avoid this problem and achieve much higer efficiency compared with StreamingLLM.
>
> **[Response]** You are correct to note that this process has an effect on inference efficiency. This can be directly observed in Figure 6a where the zoomed in subfigure shows that our cascading cache with one cascade (equivalent to Streaming LLM) has lower latency than 4 cascades. **However, we note that the difference is on the scale of 4 orders of magnitude smaller than the difference between the original Streaming LLM and our cascading cache due to the efficiency of our implementation** which is why we needed to create the zoomed in subgraph of figure 6a to even visualize it.
>
> So we believe our method in fact does achieve higher efficiency as compared with Streaming LLM, with a negligible overhead that increases with the number of cascades. We leave any further improvements on top of this as interesting lines of future work.
>
> ---
>
> **[W5]** Token selection [...] (EMA). [...] similar to H2O. Can I understand this process as considering the different importance degrees of current and historical attention score based on the method of H2O, which is controlled by the hyper-parameter gamma.
>
> **[Response]** The motivation for using an EMA comes from the fact that a sum accumulation would provide a slight bias towards keeping older tokens which have had more strictly positive terms in the sum, while an EMA would not have this bias. To compare the sum and the EMA, we evaluated a sum accumulation on the LongBench experiment which shows that the EMA has a slight advantage over a sum, although a sum still performs much better than Streaming LLM.
>
> |  | Narrative QA | HotPotQA | Qasper | Multi News | 2WikiMQA | Gov. Report | Avg. |
> |-|-|-|-|-|-|-|-|
> | Streaming LLM | 22.57 | 40.78 | 23.89 | 20.69 | 23.46 | 26.82 | 26.37 |
> | Cascade (Ours, Sum) | 25.45 | 47.24 | 32.95 | 23.11 | 31.89 | **28.53** | 31.19 |
> | Cascade (Ours, EMA) | **26.43** | **47.26** | **33.12** | **23.33** | **32.33** | 28.32 | **31.8** |
>
> ---
>
> We thank you for your time and effort in reviewing our work. We believe that that added experiments have improved the quality and thoroughness of our evaluation. If you have any remaining concerns after our response, we remain open to further discussion throughout the discussion period.

---

> > ### Comment · Reviewer_4zw7 · 2024-12-02
> >
> > Thanks for the detailed responses. I do not have further questions and I will raise my score considering all the updated materials.

---

> > > ### Author Response · Authors · 2024-12-02
> > >
> > > Thank you for your time and effort in reviewing our work. We are glad to hear your positive response. We noticed that you stated you will raise your score, however this is not yet reflected in the OpenReview score. We kindly ask that you reflect your score update in OpenReview by editing the score on the original review.
> > >
> > > Thank you,
> > > Authors

---

### Official Review · Reviewer_Ckne · 2024-11-03

**Soundness:** 3
**Presentation:** 3
**Contribution:** 3
**Rating:** 6
**Confidence:** 3

**Summary:**

The paper presents a new training-free approach which significantly extend the effective context length of LLM. The authors first introduce a strided prefill which reduces the prompt computing latency on 1M tokens by a factor of 6.8 compared to flash attention. Then illustrate their cascade KV cache maintaining mechanism. Given the same KV cache size, the cascade mechanism gains impressive improvements compared with the existing methods.

**Strengths:**

1. Although the idea is simple, the experiment results are very good.
2. The idea is well implemented with careful code optimization.

**Weaknesses:**

The experimental section would be better if the idea is evaluated on more data sets.

**Questions:**

I am wondering the perplexity results on more evaluation datasets.

---

> ### Author Response · Authors · 2024-11-19
> **Author Response**
>
> Thank you for your time and care in reviewing our work and highlighting the simplicity, optimization, and strong experimental results we provide. Please see our responses to your additional comments below with 4 new evaluations. Perplexity results utilize Llama 3.1 8B, while every other result utilizes Llama 3.1 8B Instruct.
>
> ---
>
> **[W1/Q1]** The experimental section would be better if the idea is evaluated on more data sets. I am wondering the perplexity results on more evaluation datasets.
>
> We evaluated our model vs Streaming LLM on Wikitext2 test set which was concatenated into a single input consisting of 288K tokens. We use a prefill stride of 512 tokens with cache sizes of 128, 256, 512, 1024, and 2048. We find that the results are consistent with the PG19 perplexity results which show that our model performs consistently better than Streaming LLM. Please see the results in the table below:
>
> | Cache Size | 128 | 256 | 512 | 1024 | 2048 |
> |---|---|---|---|---|---|
> | Streaming LLM | 5.98 | 5.84 | 5.69 | 5.55 | 5.45 |
> | Cascade (Ours) | **5.81** | **5.67** | **5.56** | **5.47** | **5.42** |
>
> We have also evaluated our model in terms of latency and performance against Minference [1], which is a recently published benchmark raised by other reviewers. Minference does not have a fixed memory overhead, and therefore goes out of memory for sequences over 131K on an A6000 (49GB) although the latency is competitive. Our model, however, maintains a fixed memory overhead for any context size and can easily handle 1M tokens on a single A6000.
>
> |  | 16K | 32K | 65K | 131K | 262K | 524K | 1M |
> |---|---|---|---|---|---|---|---|
> | Minference | 5.11 | 12.84 | 28.41 | 60.47 | OOM | OOM | OOM |
> | Cascade (Ours) | 9.58 | 19.71 | 40.11 | 80.84 | 164.47 | 334.17 | 665.33 |
>
> We also performed an experiment evaluating against Minference and Pyramid KV on LongBench using the same truncation setting as we used for the quadratic model in Figure 9 due to the fact that Minference utilizes dense quadratic attention for every decoding step and Pyramid KV requires a quadratic prompt. Our Cascading KV Cache model maintains superior performance on every dataset except Gov. Report which shows the smallest margin between all models and methods in Figure 9.
>
> |  | Narrative QA | HotPotQA | Qasper | Multi News | 2WikiMQA | Gov. Report | Avg. |
> |---|---|---|---|---|---|---|---|
> | Pyramid KV | 20.99 | 39.79 | 19.86 | 23.2 | 21.77 | **29.2** | 25.63 |
> | Minference | 20.9 | 39.79 | 19.77 | 23.19 | 21.52 | 29.15 | 25.72 |
> | Streaming LLM | 22.57 | 40.78 | 23.89 | 20.69 | 23.46 | 26.82 | 26.37 |
> | Cascade (Ours) | **26.43** | **47.26** | **33.12** | **23.33** | **32.33** | 28.32 | **31.8** |
>
> We also extended our experiments to InfiniteBench [2] using a total cache size of 32K and compared to the Minference, finding the same pattern that was shown in LongBench where **our model shows large improvements  of (21.5% and 18.2%) on the first two datasets**, and underperforms Minference by only 2.3% on the last dataset where the overall differences between models is much smaller.
>
> |  | en.MC | en.QA | en.Sum | Avg. |
> |-|-|-|-|-|
> | Streaming LLM | 46.72 | 13.98 | 30.8 | 30.5 |
> | Minference | 46.72 | 14.96 | **32.25** | 31.31 |
> | Cascade (Ours) | **56.77** | **17.69** | 31.5 | **35.32**|
>
> ---
>
> Thank you for your time and effort in reviewing our work. Your suggestions have improved the thoroughness of our experiments. We hope we have adequately responded to your discussion points. If you have any remaining discussion points, please let us know. Thank you.
>
> ---
>
> ## References
>
> [1] Minference: https://arxiv.org/pdf/2407.02490
>
> [2] Infinite Bench: https://arxiv.org/abs/2402.13718

---

### Official Review · Reviewer_2xPR · 2024-11-03

**Soundness:** 2
**Presentation:** 3
**Contribution:** 3
**Rating:** 6
**Confidence:** 4

**Summary:**

The transformer’s context window is crucial for tasks like few-shot learning, but longer contexts increase computational costs, limiting large language models (LLMs) in long-sequence tasks. Recent key-value caching methods lower complexity but often discard tokens too soon and have inefficient prefill stages, leading to high latency. To solve this, the author propose a cascading sub-cache buffer system that selectively retains relevant tokens, supporting longer context without added cache size. This method improves accuracy and reduces latency across tasks, enhancing LLM efficiency for real-time applications in resource-limited environments.

**Strengths:**

- The idea of cascade kv cache is simple and effective
- The design of this method carefully considered the implementation so as to get the actual speedup

**Weaknesses:**

The baseline may not be sufficient, most of the experiment are comparing with streamingLLM

**Questions:**

- How's your peformance compared with Minference?
- How's your performance compared with others on RULER task, which is a benchmark for evaluating context size

---

> ### Author Response · Authors · 2024-11-19
> **Author Response**
>
> Thank you for your time in reviewing our work and highlighting the strengths of our method being simple, effective, and fast. Please see our comments to your further responses below:
>
> ---
>
> **[W1]** The baseline may not be sufficient, most of the experiment are comparing with streamingLLM
>
> **[Response]** Streaming LLM is the key baseline for our method, as it is the only method from the relevant baselines which is truly linear in complexity with a fixed memory overhead like ours.
>
> - In our experiments, we have included Streaming LLM, BigBird, full quadratic attention, SnapKV, H2O.
> - The only experiment which compares to Streaming LLM without other baselines is the 1M token passkey retrieval which is due to the fact the the other methods have quadratic latency as shown in figure 6 which makes them infeasible for this task, and/or they require access to all tokens in the sequence (BigBird) which incurs a large overhead memory cost.
>
> **Streaming LLM and our Cascading KV cache are the only two methods which offer linear complexity with a fixed memory overhead for any sequence length.**
>
> ---
>
> **[Q1]** How's your peformance compared with Minference?
>
> **[Response]** In terms of latency, our cascading cache is competitive with Minference at different context lengths, However Minference exceeds the memory of an A6000 (49GB) after 131K context length which is the environment we ran our experiments in. The full model latency for all layers (seconds) can be seen in the table below. All results posted below utilize Llama 3.1 8B Instruct.
>
> |  | 16K | 32K | 65K | 131K | 262K | 524K | 1M |
> |---|---|---|---|---|---|---|---|
> | Minference | 5.11 | 12.84 | 28.41 | 60.47 | OOM | OOM | OOM |
> | Cascade (Ours) | 9.58 | 19.71 | 40.11 | 80.84 | 164.47 | 334.17 | 665.33 |
>
> We also performed an experiment evaluating against Minference and Pyramid KV on LongBench using the same truncation setting as we used for the quadratic models in Figure 9 due to the fact that Minference utilizes dense quadratic attention for every decoding step and Pyramid KV requires a quadratic prompt.
>
> |  | Narrative QA | HotPotQA | Qasper | Multi News | 2WikiMQA | Gov. Report | Avg. |
> |---|---|---|---|---|---|---|---|
> | Pyramid KV | 20.99 | 39.79 | 19.86 | 23.2 | 21.77 | **29.2** | 25.63 |
> | Minference | 20.9 | 39.79 | 19.77 | 23.19 | 21.52 | 29.15 | 25.72 |
> | Streaming LLM | 22.57 | 40.78 | 23.89 | 20.69 | 23.46 | 26.82 | 26.37 |
> | Cascade (Ours) | **26.43** | **47.26** | **33.12** | **23.33** | **32.33** | 28.32 | **31.8** |
>
> We find that our cascading KV cache outperforms Minference and Pyramid KV on every dataset except for Gov. Report which shows the tightest margin across all models and methods.
>
> ---
>
> **[Q2]** How's your performance compared with others on RULER task, which is a benchmark for evaluating context size?
>
> **[Response]** Compared to RULER, we believe that InfiniteBench [1] offers a more comprehensive and realistic assessment of model performance in long contexts. InfiniteBench evaluates tasks like multiple-choice reasoning, question answering, and summarization with average context lengths of 182K, 192K, and 171K, respectively. We used a total cache size of 32K. These tasks go beyond string recall, emphasizing a model's ability to extract and utilize meaningful information across large contexts—a critical capability for practical applications. We evaluate 3 datasets from InfiniteBench below:
>
> |  | en.MC | en.QA | en.Sum | Avg. |
> |-|-|-|-|-|
> | Streaming LLM | 46.72 | 13.98 | 30.8 | 30.5 |
> | Minference | 46.72 | 14.96 | **32.25** | 31.31 |
> | Cascade (Ours) | **56.77** | **17.69** | 31.5 | **35.32**|
>
> We observe the same trend as seen in LongBench, whereby our model outperforms by a large margin (21.5% and 18.2%) on the first two datasets, and underperforms Minference by only 2.3% on the last dataset where the overall differences between models is much smaller.
>
> ---
>
> Thank you for your time and effort in reviewing our work. Your thoughtful suggestions have improved the quality and completeness of our work. We hope we have adequately responded to your comments. If you have any remaining discussion points, please let us know. Thank you.
>
> ---
>
> ### References
>
> [1] Infinite Bench: https://arxiv.org/pdf/2402.13718

---

> > ### Comment · Reviewer_2xPR · 2024-12-02
> >
> > Thanks for the detailed responses. My final concern is your passkey retrieval result in your paper is not as good as that of Minference, shown in their paper, do you have any results or explanations on this benchmark?

---

> > > ### Author Response · Authors · 2024-12-02
> > > **Cascade/MInference differences**
> > >
> > > Thank you for your response. The passkey retrieval accuracy can be explained by the constraints on the total cache size. Here are some interesting points to consider:
> > >
> > > ## MInference
> > >
> > > - **MInference stores 1M tokens** in the cache for 1M token passkey retrieval.
> > > - MInference must also use a special model which is **finetuned on 1M sequence** lengths in order to even run this experiment. This is because they are always limited by the number of positional embeddings used in finetuning.
> > > - MInference prefill scales superlinearly as demonstrated in the general response above where doublings of the sequence lengths result in more than doubling of the latency.
> > >
> > > ## Ours
> > >
> > > - Our method, however, **only stores 65K (6% compared to MInference) tokens** for this experiment, and can effectively extend the pretrained positional encoding in any model. We used the official Llama3.1-8b-instruct model which was only trained with 131K positional embeddings and our method was able to operate on sequences of 1M in length.
> > >
> > > - While **our method offers a linear prefill and automatic extension of the pretrained positional embeddings**, it comes with the cost of added sparsity in the latter cascades which is a barrier to perfect passkey retrieval. This is why the accuracy slowly degrades as the passkey is inserted deeper in a very long sequence.
> > >
> > > ---
> > >
> > > While there are pros and cons to both methods, we believe our Cascading KV Cache provides an excellent and immediately applicable option which can offer linear complexity, fixed memory overhead, and extensions to context lengths far beyond those used in pretraining by only changing the cache class during inference.

---

> > > > ### Comment · Reviewer_2xPR · 2024-12-02
> > > >
> > > > I got your point; I have no further questions and will raise my score.

---

> > > > > ### Author Response · Authors · 2024-12-02
> > > > > **Thank you**
> > > > >
> > > > > Thank you for your time and effort in evaluating our work, we feel these comparisons and integrations have improved our submission.

---

### Official Review · Reviewer_hjX9 · 2024-11-03

**Soundness:** 2
**Presentation:** 2
**Contribution:** 2
**Rating:** 6
**Confidence:** 5

**Summary:**

The paper proposes an extension to attention sink by (1) blockwise prefilling to optimize the latency of this period, (2) cascading caching management. The resulting framework shows quality and latency improvement over StreamingLLM.

**Strengths:**

1. Prefilling observation is sound. It's good to bulk processing this stage, which StreamingLLM and H2O failed to do.
2. It's intuitionally beneficial to keep the middle context in addition to sliding window and attention sink, which the paper proves by passkey performance.

**Weaknesses:**

1. It's pretty tricky to manage memory in this cascading way. I would say it's challenging to manage small amount of memory, e.g., two bytes, at arbitrary locations. It's very likely to cause memory fragmentation and hurt the throughput as a result, which a practical serving system should avoid.
2. There are a few earlier and more comprehensive works (compared to StreamingLLM) that tackles the prefilling stage with sparsity, e.g., MInference[1], it would be compare against [1].
[1] MInference 1.0: Accelerating Pre-filling for Long-Context LLMs via Dynamic Sparse Attention

**Questions:**

1. Related to weakness 1, is this method compatible with continuous batching? How well does the method perform compared to, or compatible with actual serving systems, e.g., vllm/sglang?
2. Could you benchmark the latency and quality wrt to Minference?
3. For FlashAttention, which version are you comparing to? FlashAttention-1, FlashAttention-2, or FlashAttention-3? Triton or cuda kernel?
4. What would be the memory overhead, and IO overhead for computing, storing and access the EMA for each token(and manipulate the resulting memory layout change)? Also, can this operation be fused into online softmax? It would be good to add a pseudo code section to illustrate how this part is done, e.g., based on FlashAttention logic.

---

> ### Author Response · Authors · 2024-11-19
> **Author Response I**
>
> Thank you for your time and care in reviewing our work, and highlighting the strengths of our prefill method, as the prefill stage is not considered in many prior works. Please see our response to your other comments below.
>
> ---
>
> **[W1]** It's pretty tricky to manage memory in this cascading way. I would say it's challenging to manage small amount of memory, e.g., two bytes, at arbitrary locations. It's very likely to cause memory fragmentation and hurt the throughput as a result, which a practical serving system should avoid.
>
> **[Response]** Our cache allocates a contiguous memory buffer at initialization. Therefore, All sub-cache buffers are actually part of the same contiguous buffer, with boundaries being set analytically. By doing this, we mitigate any possible memory fragmentation issues. However, due to the update procedure, the stored tokens will not necessarily be in sorted order. Therefore, we also track the integer positional indices as well in order to apply the correct positional encodings.
>
> ---
>
> **[W2]** There are a few earlier and more comprehensive works (compared to StreamingLLM) that tackles the prefilling stage with sparsity, e.g., MInference[1], it would be compare against [1]. [1] MInference 1.0: Accelerating Pre-filling for Long-Context LLMs via Dynamic Sparse Attention
>
> **[Response]** Our cascading cache is optimized for scaling to extremely large contexts on a **fixed memory budget**, a scenario where Minference struggles due to its memory overhead. In terms of latency, our cascading cache is comparable with Minference at different context lengths, However minference exceeds the memory of an A6000 (49GB) after 131K context length which is the environment we ran our experiments in. The full model latency for all layers (seconds) can be seen in the table below. All additional experiments below utilize Llama 3.1 8B Instruct models.
>
> |  | 16K | 32K | 65K | 131K | 262K | 524K | 1M |
> |---|---|---|---|---|---|---|-|
> | Minference | 5.11 | 12.84 | 28.41 | 60.47 | OOM | OOM | OOM |
> | Cascade (Ours) | 9.58 | 19.71 | 40.11 | 80.84 | 164.47 | 334.17 | 665.33 |
>
> We compare Minference and Pyramid KV with our cascading method on LongBench, we truncate the inputs in the same manner as for the quadratic model in Figure 9 due to the fact that Minference utilizes full dense attention for each decoding step and Pyramid KV utilizes a quadratic prompt. We find the following results:
>
> |  | Narrative QA | HotPotQA | Qasper | Multi News | 2WikiMQA | Gov. Report | Avg. |
> |---|---|---|---|---|---|---|---|
> | Pyramid KV | 20.99 | 39.79 | 19.86 | 23.2 | 21.77 | **29.2** | 25.63 |
> | Minference | 20.9 | 39.79 | 19.77 | 23.19 | 21.52 | 29.15 | 25.72 |
> | Streaming LLM | 22.57 | 40.78 | 23.89 | 20.69 | 23.46 | 26.82 | 26.37 |
> | Cascade (Ours) | **26.43** | **47.26** | **33.12** | **23.33** | **32.33** | 28.32 | **31.8** |
>
> The only dataset where Minference outperformed our model was on Gov. Report which by far shows the narrowest gap between models across all experiments shown in figure 9 in our paper.
>
> Additionally, we added a comparison to Minference on 3 datasets from InfiniteBench [1] which evaluates tasks like multiple-choice reasoning, question answering, and summarization with average context lengths of 182K, 192K, and 171K, respectively. WE used a total cache size of 32K. Like LongBench, these tasks evaluate a model's ability to extract and utilize meaningful information across large contexts—a critical capability for practical applications.
>
> |  | en.MC | en.QA | en.Sum | Avg. |
> |-|-|-|-|-|
> | Streaming LLM | 46.72 | 13.98 | 30.8 | 30.5 |
> | Minference | 46.72 | 14.96 | **32.25** | 31.31 |
> | Cascade (Ours) | **56.77** | **17.69** | 31.5 | **35.32**|
>
> We observe the same trend as seen in LongBench, whereby our model outperforms by a large margin (21.5% and 18.2%) on the first two datasets, and underperforms Minference by only 2.3% on the last dataset where the overall differences between models is much smaller.
>
> ---
>
> Continued in next comment...

---

> ### Author Response · Authors · 2024-11-19
> **Author Response II**
>
> **[Q1]** Related to weakness 1, is [...] compatible with continuous batching? [...] compatible with actual serving systems, e.g., vllm/sglang?
>
> **[Response]** Yes, our method is compatible with continuous batching because it can be adopted into PagedAttention frameworks (vLLM, SGlang). We intend to provide these integrations in a future release of our code. If you foresee any way in which our method may be problematic with continuous batching, then please let us know for further discussion.
>
> ---
>
> **[Q3]** Which version are you comparing to? FlashAttention-1,2,3? Triton or cuda kernel?
>
> **[Response]** The models which state vanilla quadratic Flash Attention use the official FlashAttention-2 cuda kernels. For our model which utilizes FlashAttention, we use a modified FlashAttention2 Triton kernel. We have added this distinction to the text whenever referencing flash attention, and also added the distinction between Triton and cuda kernels at the end of section 4 paragraph 1.
>
> ---
>
> **[Q4]** What would be the memory/IO overhead for computing, storing and access the EMA for each token(and manipulate the resulting memory layout change)? Also, can this operation be fused into online softmax? [...] add a pseudo code section to illustrate how this part is done.
>
> **[Response]** Since the EMA is a single scalar for every item in the cache, the memory overhead would be 2 bytes per vector which is stored in the cache (the total cache size is a constant). The computation of the EMA in the inner loop consists of a simple instantiation of the vector of coefficients, and a sum over the columns of the current block of $QK^\top$ in the inner loop. The final EMA update is performed using an atomic sum.
>
> This operation can be fused into the online softmax. The main obstacle to overcome when fusing into the online softmax is the normalization constant $c$, which is not known at each block of computation in the flash attention kernel. We overcome this with a simple approximation which assumes that the remaining mass in the normalization constant is proportional to the currently accumulated mass. Concretely, if there are are 10 total KV blocks $\text{TB} = 10$ in the flash attention computation and we have already computed 5 blocks $\text{CB} = 5$, then we adjust the current normalization constant as:
>
> $$
> c^\prime = c + c \frac{\text{TB} - \text{CB}}{\text{CB}}
> $$
>
> which is equivalent to $c^\prime = c + c ((10-5)/5) = 2c$ when computing the EMA after computing block 5 in the example above.  **We have added pseudocode for this process in algorithm 3 in our updated paper.** Note that figure 3 shows the above approximation is effective compared with no token selection. Additionally, we performed a further experiment on LongBench to evaluate whether a sum over attention scores is better than the EMA. We find that the EMA performs best.
>
> |  | Narrative QA | HotPotQA | Qasper | Multi News | 2WikiMQA | Gov. Report | Avg. |
> |-|-|-|-|-|-|-|-|
> | Streaming LLM | 22.57 | 40.78 | 23.89 | 20.69 | 23.46 | 26.82 | 26.37 |
> | Cascade (Ours, Sum) | 25.45 | 47.24 | 32.95 | 23.11 | 31.89 | **28.53** | 31.19 |
> | Cascade (Ours, EMA) | **26.43** | **47.26** | **33.12** | **23.33** | **32.33** | 28.32 | **31.8** |
>
> ---
>
> Thank you for your time and effort in evaluating our work. Your suggestions have improved the quality and completeness of our work. We hope we have adequately addressed the points you raised in your review. If you have any remaining discussion points, please let us know. Thank you.
>
> ---
>
> ### References
>
> [1] Infinite Bench: https://arxiv.org/abs/2402.13718

---

### Author Response · Authors · 2024-11-27
**Integration with MInference Kernel**

Dear Reviewers,

Thank you again for your thoughtful and constructive feedback on our work. As multiple reviewers requested comparisons with MInference, we have conducted additional experiments and provided results in our previous responses.

To address these comments even more comprehensively, we have integrated the MInference attention kernel directly into our linear strided prefill strategy. This integration allows us to replace the dense leading attention window in our original method with MInference, while retaining the core striding and cascading cache strategies of our method.

Specifically, the attention operation uses the following overall flow to integrate with MInference:
  1. Retrieve tokens from cascading cache
  2. Perform attention for cached tokens + MInference window
  3. Add tokens from the leading MInference window to the cascading cache.
  4. Repeat 1-3 until entire sequence is prefilled.
  5. Perform generation using the keys and values in our cascading cache.

## MInference Integration Results

By incorporating MInference, we observe significant performance improvements across most LongBench datasets, as summarized below:

|  | Narrative QA | HotPotQA | Qasper | Multi News | 2WikiMQA | Gov. Report | Avg. |
|---|---|---|---|---|---|---|---|
| Streaming LLM | 22.57 | 40.78 | 23.89 | 20.69 | 23.46 | 26.82 | 26.37 |
| Cascade (Ours) | 26.43 | 47.26 | 33.12 | **23.33** | 32.33 | **28.32** | 31.8 |
| Cascade (Ours) + MInference | **29.84** | **55.22** | **36.51** | 23.14 | **48.85** | 28.2 | **36.96** |

This boost in performance is attributed to processing longer strides, enabling more tokens from prior steps to influence the current attention computation. For these experiments, the stride for MInference + Cascade was set to 100K, which is close to the maximum for the model (131K) minus an amount to account for the cascading cache size.

---

## Latency Comparison:

We also evaluated the per-layer latency of MInference, Cascade, and the combined method. The results are as follows:

|                             | 16K  | 32K  | 64K  | 131K | 262K | 524K  | 1M    |
|-----------------------------|------|------|------|------|------|-------|-------|
| Minference                  | 0.26 | **0.28** | **0.78** | 2.26 | 6.39 | 16.84 | OOM   |
| Minference + Cascade (Ours) | 0.34 | 0.68 | 1.3  | 3.04 | 5.77 | 11.88 | 24.13 |
| Cascade (Ours)              | **0.22** | 0.46 | 0.94 | **1.93** | **3.81** | **7.66**  | **15.34** |

These results demonstrate that while integrating MInference incurs slightly higher latency compared to our original method, it still scales linearly with context length and achieves lower latency than plain MInference above 131K—a property of interest for long-context streaming settings. Note that these latency results are for one individual self attention layer, as opposed to the full model forward which is part of the response to individual reviewers.

---

## Invitation for Further Discussion

We hope these additional results address your questions and concerns. We remain open to further discussion and would greatly value your thoughts on this updated integration during the remaining discussion period.

Thank you again for your time and insightful feedback.

Sincerely,

Authors

---

### Author Response · Authors · 2024-12-02
**Discussion period will end soon**

Dear Reviewers,

As a gentle reminder, **we only have about 30 hours left for the author/reviewer discussion period.**

We politely ask that you review and respond to our updates after your initial feedback. We look forward to addressing any remaining questions which you might have and engaging in further discussion.

We sincerely thank you for your reviews which have improved our work,
Authors

---

### Meta-Review · Area_Chair_UeLe · 2024-12-23

**Metareview:**

This paper proposes a cascading sub-cache buffer system that selectively retains relevant tokens, supporting longer context without added cache size. This method improves accuracy and reduces latency across tasks, enhancing LLM efficiency for real-time applications in resource-limited environments.

The paper presentation is clear. Extensive evaluations show the effectiveness of the proposed approach.

**Additional Comments On Reviewer Discussion:**

The authors conducted a nice rebuttal to address questions. After the rebuttal, all reviewers stay positive about this paper.

---

### Decision · Program_Chairs · 2025-01-22

Accept (Poster)